# Designing Neural Dynamics: From Digital Twin Modeling to Regeneration

**DOI:** 10.3390/ijms27010122

**Published:** 2025-12-22

**Authors:** Calin Petru Tataru, Adrian Vasile Dumitru, Nicolaie Dobrin, Mugurel Petrinel Rădoi, Alexandru Vlad Ciurea, Octavian Munteanu, Luciana Valentina Munteanu

**Affiliations:** 1Department of Opthamology, “Carol Davila” University of Medicine and Pharmacy, 020021 Bucharest, Romania; 2Puls Med Association, 051885 Bucharest, Romania; 3Central Military Emergency Hospital “Dr. Carol Davila”, 010825 Bucharest, Romania; 4Department of Pathology, Faculty of Medicine, “Carol Davila” University of Medicine and Pharmacy, 030167 Bucharest, Romania; 5“Nicolae Oblu” Clinical Hospital, 700309 Iasi, Romania; 6Department of Neurosurgery, “Carol Davila” University of Medicine and Pharmacy, 050474 Bucharest, Romania; 7Department of Vascular Neurosurgery, National Institute of Neurology and Neurovascular Diseases, 077160 Bucharest, Romania; 8Medical Section, Romanian Academy, 010071 Bucharest, Romania; 9Neurosurgery Department, Sanador Clinical Hospital, 010991 Bucharest, Romania; 10Department of Anatomy, “Carol Davila” University of Medicine and Pharmacy, 050474 Bucharest, Romania

**Keywords:** attractor landscapes, adaptive neurocontrol, nonlinear brain dynamics, glymphatic–venous coupling, astrocytic signaling, epigenetic neurorepair, closed-loop neuromodulation, digital twin neuroscience, AI-driven brain modeling, systems neuroengineering

## Abstract

Cognitive deterioration and the transition to neurodegenerative disease does not develop through simple, linear regression; it develops as rapid and global transitions from one state to another within the neural network. Developing understanding and control over these events is among the largest tasks facing contemporary neuroscience. This paper will discuss a conceptual reframing of cognitive decline as a transitional phase of the functional state of complex neural networks resulting from the intertwining of molecular degradation, vascular dysfunction and systemic disarray. The paper will integrate the latest findings that have demonstrated how the disruptive changes in glymphatic clearance mechanisms, aquaporin-4 polarity, venous output, and neuroimmune signaling increasingly correlate with the neurophysiologic homeostasis landscape, ultimately leading to the destabilization of the network attraction sites of memory, consciousness, and cognitive resilience. Furthermore, the destabilizing processes are exacerbated by epigenetic silencing; neurovascular decoupling; remodeling of the extracellular matrix; and metabolic collapse that result in accelerating the trajectory of neural circuits towards the pathological tipping point of various neurodegenerative diseases including Alzheimer’s disease; Parkinson’s disease; traumatic brain injury; and intracranial hypertension. New paradigms in systems neuroscience (connectomics; network neuroscience; and critical transition theory) provide an intellectual toolkit to describe and predict these state changes at the systems level. With artificial intelligence and machine learning combined with single cell multi-omics; radiogenomic profiling; and digital twin modeling, the predictive biomarkers and early warnings of impending collapse of the system are beginning to emerge. In terms of therapeutic intervention, the possibility of reprogramming the circuitry of the brain into stable attractor states using precision neurointervention (CRISPR-based neural circuit reprogramming; RNA guided modulation of transcription; lineage switching of glia to neurons; and adaptive neuromodulation) represents an opportunity to prevent further progression of neurodegenerative disease. The paper will address the ethical and regulatory implications of this revolutionary technology, e.g., algorithmic transparency; genomic and other structural safety; and equity of access to advanced neurointervention. We do not intend to present a list of the many vertices through which the mechanisms listed above instigate, exacerbate, or maintain the neurodegenerative disease state. Instead, we aim to present a unified model where the phenomena of molecular pathology; circuit behavior; and computational intelligence converge in describing cognitive decline as a translatable change of state, rather than an irreversible succumbing to degeneration. Thus, we provide a framework for precision neurointervention, regenerative brain medicine, and adaptive intervention, to modulate the trajectory of neurodegeneration.

## 1. Introduction—From Static Descriptions to Dynamic Control of Brain States

Neuroscience has grown over the last 100+ years primarily as descriptive (how do neurons encode info; how do networks create cognition; how do molecular programs guide synaptic plasticity). Descriptive research has given us great insights into the cellular substrates and structural correlates of the functions of the brain. However, because the primary way researchers have thought of the brain has been as a “map” of the brain’s structures rather than as an evolving system that is active over time, most of our current treatments for neurological and psychiatric illnesses rely on treating damaged structures or suppressing symptomatology [1]. Many significant clinical events—seizure emergence; chronic depression; failure to recover consciousness following traumatic brain injury (TBI)—cannot be explained by damage to static brain structures alone, suggesting that what is being lost is the ability to move appropriately through adaptive brain states [2].

The brain is a complex, nonlinear dynamical system that operates across a variety of spatial and temporal scales. Therefore, the best way to conceptualize its function is as a trajectory in a high-dimensional state space that is determined by the excitability of neurons; the connectivity of synapses; the signaling of molecules; the dynamics of glia; and vascular–metabolic relationships [3]. Stable patterns of coordinated activity emerge as attractors toward which neural trajectories tend to converge. Canonical modes of activity, such as sleep, attention, working memory, mood regulation, and consciousness, are examples of attractors, and pathological regimes such as those seen during an epileptic seizures can also serve as attractors [4]. Health depends on two things: the stability of attractors and the flexibility to transition between different basins of attraction based on internal goals and external demands. Transitions between basins are regulated by a set of interacting mechanisms: the excitatory–inhibitory balance; activity-dependent synaptic weighting; the tone of neuromodulators; the signaling of calcium by astrocytes; and the regulation of blood flow and metabolism by the nervous system [5]. Therefore, pathology can be viewed as either the distortion of the state space or the loss of transition capacity. Seizures can occur as the result of a sudden and catastrophic bifurcation in state space, forcing the system from a normal physiological basin of attraction into a deep pathological attractor with hypersynchronized oscillations—this can happen due to relatively small perturbations in the excitatory–inhibitory balance [6]. Depression may represent the trapping of cortico-subcortical systems in rigid, low-fidelity attractors that are caused by abnormalities in the serotonergic/dopaminergic modulation; dysfunctional astrocyte signaling; abnormal glutamate–GABA cycling; and abnormalities in synaptic plasticity [7]. Traumatic brain injuries and coma can cause disruptions in the thalamo-cortical and mesocircuit dynamics, limiting access to attractors that allow for the experience of conscious processing [8]. Neurodegenerative diseases can lead to the progressive degradation of synaptic, glial, vascular, and metabolic scaffolding, reducing the number of possible states available and causing collapse of transition routes. A common theme among all the disorders mentioned here is that disease represents maladaptive trajectories in a distorted dynamical landscape—not just damaged components. The brain is unable to enter; stay in; or exit attractors necessary for functioning normally [9]. A dynamic view of the brain has immediate implications for therapy. Since illness represents the inability to make transitions between states, the process of recovery involves the reconfiguration of the topography of attractors and the steering of trajectories toward adaptive regimes. To accomplish this will require three related abilities: creating maps of individualized state spaces; predicting where trajectories will evolve in the short term; and intervening at the correct time and scale. Three areas of science are beginning to converge to create these capabilities: digital twins; adaptive control systems; and precision neurotherapeutics [10].

Digital twins are individualized, dynamic representations of brains that incorporate a wide range of data types (e.g., imaging, electrophysiology, genetics) that capture the brain’s structure and function. As new data become available, the model is continuously updated. While fully integrating data across multiple scales (from single cell multi-omics to behavior) in real time is currently a future direction, there are already several applications of digital twins that enable clinically important tasks (e.g., personalized connectome mapping; predicting when someone is likely to have a seizure; and predicting recovery after TBI) [11,12]. Twins transform the concept of disease from a static category to an altered dynamical structure and provide a platform for exploring how alterations in the system may redirect trajectories [13].

Adaptive control provides the mechanism for steering. Control theory provides the theoretical framework for how to drive complex systems toward specific states under constraints of feedback, stability, and estimation. For neuroscience, controllability, observability, and closed-loop regulation are becoming increasingly important concepts [14,15]. Examples of adaptive control in clinical use include adaptive deep brain stimulation that adjusts the amount of stimulation provided to the subthalamic nucleus based on changes in subthalamic activity to minimize pathological beta oscillations in Parkinson’s disease; responsive neurostimulation that detects the onset of preictal dynamics before a seizure occurs to prevent it; and closed-loop transcranial magnetic stimulation (TMS) that maintains cortical excitability within a predetermined range of adaptability [16]. Predictive controllers that utilize machine learning (such as reinforcement learning) attempt to predict the onset of pathological transitions and intervene proactively. These approaches are in their infancy and must address issues such as nonstationarity of biological processes; latency; and safety ceilings—however, they establish a potential pathway for the proactive steering of trajectories [17]. However, even if we have developed methods for steering, without changing the underlying landscape of the brain, we will continue to be limited in terms of how much we can achieve. Precision neurotherapeutics extends control into molecular and cellular parameters that shape the geometry of attractors: the distribution and kinetic properties of ion channels; the rules governing synaptic/structural plasticity; the signaling of astrocytes/microglia; the tone of the neuroimmune system; and the gradient of neuromodulators [9].

Increasingly sophisticated tools exist to modify these parameters with greater specificity. Genome and epigenome editing via CRISPR technology allows for the modification of plasticity programs and neuromodulatory pathways and may potentially open up previously inaccessible critical period like windows or alter the timing and kinetics of bifurcations and transitions [18]. Altering the kinetic properties of potassium/chloride channels can modify excitability landscapes, and modifying the localization of aquaporin-4 in astrocytes can affect ionic and metabolic homeostasis relevant to attractor stability [19]. Chemogenetic/optogenetic actuators and time-locked pharmacology enable selective, state-aligned interventions. When used in conjunction with predictive modeling guided by digital twins and adaptive control, these techniques may enable multiscale modification of the brain—from ion channel microphysics to behavior. However, the majority of the durable reprogramming of gene expression remains preclinical and requires extensive validation [20].

In combination, the development of individualized modeling, adaptive control, and molecular precision enables an emerging paradigm: not only to understand or repair the brain but also to engineer state transitions towards stable adaptive regimes. In this context, pathologies are distortions in state space geometry, and therapies are intended to reorganize the dynamical landscape of the brain, rather than simply reduce symptoms [21,22]. The clinical question moves from correcting a lesion or neurotransmitter level to enabling the system to navigate to a robust and flexible dynamical regime [23]. An example of the proof-of-principle of this approach includes individuals who suffer from refractory epilepsy treated with a combination of individualized seizure prediction and closed loop stimulation; recovery after TBI supported by the generation of stimulation plans that facilitate access to conscious attractors; and individuals with neurodegenerative diseases treated with combinations of neuromodulatory and epigenetic strategies designed to preserve or increase the number of possible states available to them [24]. In each of these cases, treatment is an iterative dialogue with endogenous plasticity and facilitates reorganization and adaptation [25].

Despite the advances in this field, there are a number of significant challenges that remain. First, the development of digital twins is constrained by the availability of large amounts of high-dimensional, noisy, and partially sampled data. Second, the implementation of adaptive control algorithms must remain stable and safe in the presence of biological variability and the risk of inducing maladaptive feedback. Finally, molecular interventions will be subject to various forms of compensation from homeostatic loops; delivery barriers; long-term safety concerns; and regulatory scrutiny [26]. There are also a number of ethical and legal considerations that must develop to protect neural data privacy; define accountability for closed loop decision making; ensure equitable access; and limit the unregulated manipulation of cognition and/or identity [27].

Rather than decreasing the need for interdisciplinary collaboration across systems neuroscience, control theory, molecular biology, artificial intelligence (AI) and clinical practice, the challenges outlined above emphasize the need for continued and enhanced interdisciplinarity.

This paper outlines a conceptual structure for transitioning brain states through engineering. This conceptualization of neurological illness as resulting from aberrant trajectory in distorted state space examines how digital twins allow individualized determination and forecasting of states, how adaptive control allows one to guide these trajectories; and how precision neurotherapeutic interventions may alter the topological properties of an attractor to restore the ability to transition between states. The purpose of our synthesis is to establish a relationship among mechanisms ranging from molecular to systems and provide translational direction that is grounded in evidence while acknowledging present limitations and avenues for testing.

## 2. The Brain as a Dynamical Landscape—Attractors, Criticality, and Molecular Underpinnings

Classical Neuroscience views the brain as hierarchical circuits, cells, and molecules that can be mapped and measured. However, this is just one aspect of how brains work [28]. The brain can also be seen as a non-linear dynamic system that evolves in a high-dimensional state space of electrical, molecular, cellular, metabolic, and network variables [29]. As such, neural trajectories evolve in this high-dimensional state space and converge to an attractor—a stable or metastable regime that corresponds to a physiological or cognitive state (attention, memory, sleep, arousal, mood, consciousness). Distortion of the attractor’s geometry and accessibility leads to pathological regimes (seizures, coma, depressive inertia) [30].

The constructs listed below describe a single multiscalar state-transitions framework that Section 3, Section 4, Section 5 and Section 6 will instantiate.

### 2.1. State-Space Architecture, Attractors, and Critical Transitions

The evolution of brain activity is described by the time-dependent behavior of a state vector under the influence of internal interactions, slowly changing parameters, and external inputs; the attractors are defined as fixed points, limit cycles, or recurrent/chaotic regimes toward which trajectories tend [31]. Stability is defined by whether perturbations cause the trajectory to return to its original regime, while state transitions occur at bifurcations due to gradual changes in parameters leading to qualitative changes in regimes (from asynchronous firing → synchronized oscillations; from wake → sleep; from physiological → preictal). The true state space is extremely large, but the trajectories of interest typically exist in low-dimensional manifolds bounded by structural connectivity, recurrent patterns, and neuromodulatory fields; the topology, dimensionality, and curvature of these manifolds define the rate of transition and the sensitivity of trajectories to perturbations [32,33,34]. Therefore, disease can be considered a deformation of these manifolds where there are either deeper or more accessible pathological attractors and a contraction of the range of stable states [35].

Operational descriptions of the geometry of the manifolds can be obtained using Topological Data Analysis. Persistent Homology describes the invariants (such as repeated loops) of the geometry of the dynamical repertoire of the system, and their loss indicates a reduction in diversity of the dynamical repertoire and an increase in the inefficiency of the transitions between regimes in disease [36]. The same manifold is used to describe the control theoretic notion of observability, i.e., whether the latent states of the system can be reconstructed from data such as electrocorticography (ECoG) or calcium imaging, and controllability, i.e., whether the input to the system (stimulation, drug effects) can move the system from one basin to another. The use of local linearization around each point of the manifold allows Gramian-based estimation of the transition energy, enabling the identification of nodes, frequencies, or paths that are likely to be good candidates for adaptive control [37,38].

Critical boundaries between different regimes are typical of many brain networks, and criticality provides scale-free organization of the network, maximum sensitivity to small variations in parameters, and maximum efficiency of information transfer [39]. When approaching a critical point, recovery of a perturbed state slows (critical slowing down), flicker between basins increases, and increasing autocorrelation or variance serve as early warning indicators of transition proximity [40].

Small changes in ion conductance, synaptic efficacy, neuromodulatory tone, or in the concentration of ions outside the cell can lead to saddle-node, Hopf, or SNIC-like bifurcations that transform physiological rhythms into pathological attractors [41]. For example, the progressive decrease in barrier height to the basins due to an increase in the extracellular concentration of potassium (K^+^) or a decrease in inhibitory action in the case of epilepsy can lead to bifurcations that trigger seizures [42].

Stochasticity is partially functional: random fluctuations in the opening/closing of ion channels and the probabilistic release of vesicles facilitate Kramers-type basin escape and stochastic resonance and allow for flexible exploration of the state space [43,44]. Homeostatic plasticity (synaptic scaling, modulation of intrinsic excitability, activity-regulated transcription) acts to counteract drift and maintain near-critical adaptability without loss of stability [45].

### 2.2. Multiscale Molecular and Cellular Shaping of the Landscape

The geometry of the attractors is determined by the interaction of fast and slow variables at multiple scales. Fast variables at the microscale (ion-channel gating, synaptic transmission) determine the moment to moment structure of the basins; medium-scale variables (network connectivity, effective connectivity, oscillatory coordination, E/I balance) determine the local curvature of the manifold; and long-scale variables (repertoire richness, behavioral performance, metastability) determine the overall regime [46]. Glial–immune tone, vascular–glymphatic clearance, metabolic state, ECM remodeling, and epigenetic programs all act as slow modulators to reorganize the basins and manifolds at longer timescales and modify the likelihood of future transitions. At the single neuron level, ensembles of ion channels define the surface of nonlinear excitability: rapid inward Na^+^ currents and slower outward K^+^ currents define the threshold for spiking and the duration of the refractory period [47].

Sub-threshold conductances (HCN, T-type Ca^2+^, KCNQ2/3 M-type K^+^) determine the resonance frequency of the membrane potential and the degree of synchronization among neurons, while phosphorylation, auxiliary subunits, lipid micro-domains, and redox state modify the curvature of the basins and the propensity for bifurcations [48,49]. Synaptic processing modifies the location and depth of the basins: presynaptic Ca^2+^ dynamics and the kinetics of the vesicle pool modify the temporal filtering and gain of the short term synaptic response, and LTP/LTD modify the strength of recurrent connectivity and thus the separation of the attractors and the width of the basins [50].

The spatial relationship between the release sites of the presynaptic neuron and the postsynaptic receptor domains defines the variability in the quantal response and the ability of the neuron to detect coincident synaptic events, thereby defining the eigenmodes of the circuit. Finally, the excitatory–inhibitory balance, established through glutamatergic/GABAergic transmission and regulated by KCC2-dependent chloride homeostasis, determines the energy barriers between the basins and the stability of the attractors [51]. Astrocytes and neuromodulatory systems act as strong slow modulators of neural dynamics. Astrocytic Ca^2+^ waves mediate gliotransmission (ATP, D-serine, glutamate), Kir4.1 buffer extracellular K^+^, AQP4 regulate the movement of water and ions, and EAATs limit the diffusion of glutamate that can accumulate in the extracellular space and deepen pathological basins [52]. Additionally, the astrocyte–neuron lactate shuttle connects synaptic demand to metabolic supply and sets the upper limit of the energy available for network states. Microglia shape the topology of the network by removing weak connections via complement-mediated pruning (C1q, C3) and cytokines, and oligodendrocytes modulate the timing of recurrent loops by modifying the speed of conduction via myelin thickness and the distance between internodes [53]. Dysfunction in these glial mechanisms (AQP4 depolarization, impaired K^+^ clearance, cytokine imbalance, increased complement activation) reduce the robustness of the network and direct trajectories toward pathological attractors. Neuromodulatory systems establish slower global biases: Dopamine establishes a bias for cortico-striatal attractors via D1/D2 receptors; Acetylcholine sharpens the transitions between attentional basins and enhances the signal to noise ratio; Noradrenaline modulates the intrinsic excitability of the neurons and regulates exploration–exploitation trade-offs; Serotonin establishes a bias for emotion related limbic and cortical attractors through various 5-HT receptors [54,55,56,57].

On even longer timescales, chromatin remodeling, histone acetylation/methylation, enhancer–promoter looping, and m6A RNA methylation regulate the gene programs for ion channels, synapses, and signaling proteins and encode previous dynamics as durable constraints on the landscape [58].

Landscape sculpting occurs on layered timescales: fast variables (ms–s) define the immediate basin boundaries and synchrony through dendritic integration, phase response curves, and conduction delays [59], intermediate variables (s–min) such as receptor trafficking, phosphorylation cycles, astrocytic Ca^2+^ signaling, extracellular K^+^/glutamate clearance, and neurovascular coupling define the depth/width of the basins and the transition probability [60,61], slow variables (h–w) including structural plasticity, myelin remodeling, ECM reorganization, epigenetic reprogramming, mitochondrial/redox dynamics, and glymphatic–vascular clearance define the geometry of the manifold and the repertoire size [62,63].

mTOR, CaMKII, and MAPK/ERK convert transient activity into CREB-stabilized connectivity, while metabolic and vascular plasticity support the preservation of the energy and ionic boundary conditions that prevent pathological basin expansion. Feedback loops between scales of biology, e.g., the cytokine effects of microglia on astrocytic AQP4 polarization and Kir4.1 buffering, modify the thresholds for bursting and the stability of the attractors [64,65,66,67].

### 2.3. Pathological Deformation and a Dynamical Basis for Intervention

Disease can be generalized as deformations of the attractor topology and manifold geometry. Ion channel mutations, interneuron dysfunction, and impaired extracellular K^+^ clearance (Kir4.1, AQP4) deepen the basins and narrow the funnel toward hypersynchronous attractors in epilepsy [68]. decreased expression of KCC2 lowers intracellular Cl-, reducing GABAergic hyperpolarization and stabilizing epileptogenic basins [69,70].

Impaired dopaminergic/noradrenergic tone, chronic inflammatory and astrocytic metabolic dysfunction, and dysregulated tryptophan–kynurenine signaling flatten the motivational manifolds and direct trajectories toward rigid low-energy attractors in depression; reduced modal controllability in limbic–prefrontal circuits increases the energy cost of exiting the maladaptive affective basins, promoting rumination and inertia [71].

Progressive removal of synaptic and white matter scaffolding, impairment of neurovascular and glymphatic regulation, collapse of manifold dimensionality, and shrinkage of accessible repertoires characterize neurodegenerative diseases [72,73,74]. After traumatic brain injury, disorders of consciousness follow disruptions in the thalamo-cortical/mesocircuit manifolds and white matter damage that alter the eigenmodes and controllability of the network, restricting access to conscious attractors. Neurodegenerative diseases result in the loss of synaptic and white matter scaffolding, impaired neurovascular/glymphatic regulation, collapse of manifold dimensionality, and shrinkage of the repertoire [75].

The primary mode for cell-to-cell signaling between neurons, astrocytes, and vascular endothelial cells (VECs) is through extracellular vesicle (EV) release. EVs contain numerous protein components that have been implicated in the progression of neurodegenerative disease including aggregation prone proteins such as amyloid-beta, phosphorylated-tau, and alpha-synuclein, as well as bioactive lipids, cytokines/complement factors, and regulatory RNA molecules [76]. These molecular cargo elements allow for the prion-like transfer of toxic seeds across connected networks and facilitate the long-term reprogramming of neuronal and glial states. Within the context of the attractor landscape model, EVs act as a form of mesoscale signal that increases the coupling of pathological variables between regions by increasing the propagation of proteostatic/mitochondrial/inflammatory stress, thus accelerating synaptic dysfunction, unbalancing excitatory/inhibitory ratios, and promoting maladaptive pruning, all which increase the depth of degenerative basins and decrease the dimensionality of state space [77].

In addition to their role in the progression of neurodegenerative diseases, EV dynamics may also contribute to clearance failure; normal functioning of the glymphatic-perivascular system would remove soluble interstitial waste, including vesicular waste. Thus, impairment of the glymphatic–venous pathway linked to aquaporin-4 is likely to prolong the presence of EVs, enhance the effect of seeding and inflammation, and increase the likelihood that a trajectory will be attracted to a rigid, low repertoire basin of attraction [78].

Across disorders, similar dynamical signatures include deep pathological basins, reduced repertoire size, loss of criticality, and inefficient transitions—measurable through critical slowing, increases in autocorrelation/variance, decreases in topological invariants, and reductions in controllability. These are the multiscale variables that digital twins will estimate and adaptive control will target in the subsequent section [79,80].

As such, therapy targets topology rather than isolated components: ion channel modulation shifts the bifurcation thresholds; selective microcircuit stimulation adjusts the excitatory/inhibitory balance; astrocytic, metabolic, and vascular interventions repair the ionic/energetic boundary conditions and normalize the depth/width of the basins; ECM modification opens up new plasticity routes; and epigenomic/epitranscriptomic interventions expand the space of slow variables and stabilize the adaptive attractors [81]. These principles motivate individualized digital twin reconstructions, adaptive control to steer trajectories, and precision neurotherapies that act on molecular parameters that define the geometry of the landscape [82,83,84]. Table 1 presents multiscale mechanisms for generating neural landscapes and developing methods for enlarging adaptive basins, contracting pathological basins, expanding repertoires, and steering trajectories towards healthier regimes.

In conclusion, brain function is the description of trajectories in a high-dimensional attractor landscape shaped by ion channels, synapses, glia, neuromodulators, epigenetic programs, metabolism, and structure. Function near a critical boundary allows for stochastic exploration of the state space stabilized by homeostatic controls. Disease causes deformations of attractor topology and manifold geometry, creating deeper pathological basins, collapsing repertoires, and increasing the energy cost of controlling transitions. The crossing of basin boundaries or the modification of basin geometry due to the interaction of fast activity variables and slowly changing modulators enables a transition toward predicting and designing brain-state transitions in health and disease [85].

**Table 1 ijms-27-00122-t001:** Multiscale mechanisms sculpting the brain’s attractor landscape and their therapeutic leverage.

Mechanistic Layer	Core Components and Processes	Contribution to Attractor Dynamics	Distortion in Disease	Therapeutic Opportunities and Emerging Interventions	References
Ion channel ensembles	Voltage-gated Na^+^, K^+^, Ca^2+^ channels; HCN; KCNQ2/3	Set intrinsic excitability and bifurcation points; tune basin curvature, resonance, and spike timing	Channelopathies reduce transition barriers and stabilize pathological basins (esp. epilepsy); altered subthreshold conductances impair critical dynamics	Channel-selective modulators to rebalance excitability; closed-loop stimulation timed to bifurcation geometry/eigenmodes	[47,86,87]
Synaptic plasticity and scaling	AMPA/NMDA trafficking; TNF-α pathways; presynaptic Ca^2+^ kinetics	Reposition and deepen/flatten basins via LTP/LTD; maintain stability through homeostatic scaling; shape local eigenmodes	Failed scaling flattens landscape and raises noise susceptibility; biased LTP/LTD shifts trajectories toward maladaptive attractors	TNF-α modulators; BDNF–TrkB agonists; metaplasticity-guided stimulation/rehab to reset transition rules	[88]
Excitation–inhibition (E/I) balance	GABA interneurons; PV circuits; KCC2/NKCC1 transporters	Determines attractor count and barrier height; stabilizes phase hierarchies and oscillatory regimes	E/I drift produces hypersynchrony or fragmentation, shrinking accessible state repertoire	KCC2 enhancement/NKCC1 tuning; interneuron grafting; gamma entrainment for targeted inhibitory control	[50,89]
Astrocytic control	AQP4; Kir4.1; EAAT1/2; connexin-43; astrocytic Ca^2+^ waves	Buffers ions and water, sets excitability thresholds; limits glutamate spillover; couples metabolism to activity	AQP4 depolarization, weak K^+^ buffering, and glutamate spillover deepen pathological basins and destabilize normal attractors	AQP4 repolarization; Kir4.1 upregulation; lactate-shuttle support to restore ionic/metabolic stability	[19,90]
Microglial–immune feedback	TREM2; APOE; complement C1q/C3; cytokines	Reshapes topology through pruning and inflammatory gain control; alters basin boundaries	Chronic inflammation stiffens boundaries and amplifies noise; over-pruning reduces controllability	TREM2 agonism; complement dampening; cytokine-targeted anti-inflammatory therapy	[91,92]
Oligodendrocytes and myelin plasticity	Myelin thickness; internode length; activity-dependent myelination	Adjusts conduction delays and synchrony, stabilizing recurrent loops and large-scale attractors	Demyelination disrupts timing, collapses coordination, narrows the dynamical repertoire	Remyelination/OPC-activation strategies; experience-driven myelin training	[93,94]
Neuromodulatory systems	Dopamine (D1/D2), acetylcholine, norepinephrine, serotonin	Provide slow global bias fields; regulate exploration vs. exploitation and state switching	Blunted modulation traps circuits in low-energy attractors (e.g., depression); unstable tone destabilizes transitions	Targeted neuromodulators (L-DOPA, nicotinic agonists, SSRIs); phase-specific stimulation of modulatory nuclei	[95]
Epigenetic and epitranscriptomic programs	Histone acetylation/methylation; enhancer looping; m6A RNA methylation	Control slow variables: stabilize new regimes, reopen/close plasticity routes, encode long-term state history	Disordered chromatin regulation limits attractor diversity and locks maladaptive states	HDAC inhibitors; m6A pathway modulators; CRISPR epigenome editing	[96,97]
Metabolic and mitochondrial dynamics	PGC-1α; SIRT3; mitophagy; lactate shuttle	Set energetic ceilings for high-amplitude states and recovery speed after perturbation	Energy failure deepens pathological basins, prolongs recovery, and drives instability	NAD+ boosters; mitophagy enhancers; metabolic-coupling interventions	[98,99]
Vascular and glymphatic support	Neurovascular coupling; pericytes; arterial pulsatility; AQP4	Sustain supply–demand matching and clearance, preserving adaptive basin geometry	Hypoperfusion/clearance failure compresses state space and increases instability	Focused ultrasound; NO-signaling support; sleep-linked clearance modulation	[100,101]
Global dynamical signatures	Criticality; bifurcations; stochastic resonance; noise	Enable high sensitivity, flexibility, and low-energy transitions	Loss of criticality reduces computational range; noise becomes destabilizing	Early-warning monitoring (critical slowing, variance, autocorrelation); resonance/phase-locked stimulation	[102]

This table synthesizes the molecular, cellular, and systems-level processes that construct and regulate the brain’s dynamical landscape, shaping the emergence, stability, and accessibility of attractor states. Each layer—from ion channel ensembles and synaptic scaling to glial regulation, neuromodulatory tone, metabolic support, and vascular dynamics—contributes distinct structural and functional roles that determine how neural trajectories evolve through state space. Disruptions in these mechanisms deform the landscape, deepening maladaptive basins, collapsing manifolds, and narrowing the repertoire of reachable states, as observed across disorders from epilepsy and depression to neurodegeneration.

## 3. Digital Twins: Building Multi-Scale Computational Mirrors of the Brain

A digital twin, within the multiscale-state transition framework described in Section 2, represents the individualized estimator for the continuously evolving brain state vector and attractor landscape. In essence, the digital twin (1) estimates the rapidly changing fast activity variables and mesoscale network coordinates through multiple modalities of data; (2) infers the slowly changing modulators that alter the shape of the basins on time scales greater than those of the activity variables; (3) identifies early warning signs that are indicative of nearness to boundaries of the attractors; and (4) forecasts how candidate inputs will influence future trajectory movements or potentially change the landscape. Therefore, the concept of a digital twin is presented in this paper not as an independent topic but rather as a necessary tool to visualize and predict state transitions in individual subjects.

Brain activity can be thought of as the motion of a complex dynamical object through a high-dimensional space. This is analogous to digital twinning—a computational representation of the same complex dynamical object. A digital twin represents an individual’s latent brain state as reconstructed from multiple sources of data, predicts the trajectory of the brain over time, and simulates potential interventions before they occur. Digital twins extend computational neuroscience from being descriptive and population-based to predictive and patient-specific, and further to adaptive and actionable [11,103]. Below, we outline the three major conceptual pillars of the digital twin: (i) adaptive dynamical mirroring via data assimilation, (ii) multiscale integration and personalization, and (iii) prediction, simulation, and realistic translational roles.

### 3.1. From Static Descriptions to Adaptive Dynamical Mirrors

Traditionally, computational models of brain function (biophysically detailed, network-based, and machine learning-based) have made important contributions to our understanding of basic principles, yet these models are generally fixed in their architecture, trained on aggregate data and separated from the individual, dynamic brain that they seek to model [104,105]. Digital twins bridge this gap by functioning as adaptive dynamical mirrors. Data from the living brain are assimilated in real time to generate probabilistic predictions of the underlying brain state and indicate proximity to attractor boundaries or impending bifurcations. They can evaluate hypothetical interventions and assess the likelihood of success of those interventions [106]. In the language of Section 2, this means estimating current basin occupancy, quantifying transition risk, and identifying whether imminent shifts reflect fast-variable instability or slow-variable landscape drift. At the heart of all digital twins is the problem of state-space reconstruction under uncertainty. Brain data (electrophysiology, imaging, biomarkers, etc.) are viewed as noisy projections of some unknown, higher-dimensional dynamical states. An observation function links these hidden states to observable data. The twin’s primary function is to infer the latent trajectory of the brain and key parameters that explain the incoming data, quantify uncertainty about the inferred trajectory, and then project the system forward in time to predict how the trajectories will evolve with or without an intervention. To achieve this goal, the twin relies on advanced families of data-assimilation algorithms: extended, unscented, and ensemble Kalman filters; particle filters; variational Bayesian inference; and sequential Monte Carlo, which fuse high-dimensional data with prior information about the dynamics of the system, producing uncertain forecasts [107]. These algorithms convert raw measurements into estimates of the evolving state of the system that can demonstrate attractor drift or pre-bifurcation instability, even if there appears to be stability in the directly observed measures [108].

A second major pillar is the use of hybrid mechanistic–machine learning models that preserve physiological knowledge and constraints while capturing nonlinearities not captured by first principles. Physics-informed neural networks (PINNs) embed constraints on differential equations (e.g., ion channel kinetics or synaptic rule) into learning objectives to restrict the possible solutions to known neurobiology while allowing the discovery of missing nonlinear structures [109]. Graph neural networks (GNNs) encode the connectome (the physical structure of the brain) and the connection strength between different parts of the brain; transformers capture long-range temporal dependencies; and variational autoencoders compress high-dimensional neural data into lower-dimensional latent manifolds that approximate the geometric structure of the attractors [110]. Thus, these are not simply black-box predictors but rather constrained dynamical system models that incorporate physiological knowledge and adaptively update themselves in response to changing aspects of the organism [111].

### 3.2. Multiscale Integration and Personalization: Twins That Co-Evolve with the Brain

Behavior is the result of multiple levels of biological organization and thus a useful digital twin must integrate variables across multiple scales (molecular, cellular, circuit, systems, and behavioral) so that perturbations at one level influence the other levels in a biologically plausible manner [112]. Here, each scale is included only insofar as it contributes measurable coordinates of the state vector or slow modulators that deform attractor geometry and bias future transitions.

At the molecular scale, twins may include kinetic equations for ion channels, the distribution of receptors (e.g., AMPA/NMDA), second messenger cascades (Ca++, cAMP), transcriptional programs, epigenetic/epitranscriptomic states (DNA methylation, chromatin accessibility, RNA methylation), and remodeling of the molecular substrates of the disease or treatment [113,114].

The near-term goal is not comprehensive molecular resolution but rather a reasonable inclusion of molecular variables that significantly limit excitability, plasticity, and threshold transition. At the cellular and microcircuit scale, twins develop nonlinear dynamical models of interacting populations of neurons and glia, incorporating short- and long-term synaptic plasticity, dendritic integration, astrocytic Ca++ signaling, microglial cytokine feedback, and adaptive myelination. These models are increasingly grounded and updated using high-density electrophysiology, two-photon Ca++ imaging and optogenetics, as these are available, to refine the local geometric structure of the attractors, separatrix location, and transition probability [115].

At the meso and network scales, twins couple structural and functional connectivity into graph-structured dynamical systems. Diffusion MRI and tractography provide anatomical constraints; fMRI, MEG, and ECoG provide functional dynamics; dynamical causal modeling infers directional connections; and network control theory identifies efficient control nodes. Using dimensionality reduction tools (Gaussian Process Factorization, Topology Data Analysis), twins can recover latent manifolds on which large-scale trajectories move through cognitive and pathological sub-spaces and thus learn how large-scale dynamical paths change with the transition between cognitive and pathological states [116,117].

Finally, at the systems-behavioral level, twins use hierarchical Bayesian models to infer latent cognitive states and reinforce learning models to link neural dynamics to action selection and adaptation, completing the loop from neural state to behavior to environment and back. Twins can now predict behavioral outcomes of neural perturbations and the neural consequences of structured behavioral or environmental changes [118,119].

Personalization is required because each individual occupies a unique region of state space due to their genetic background, developmental history, life experiences, and environment. Each disease trajectory will thus differ greatly among patients within the same diagnostic category. Twins must therefore be developed on a per-patient basis and be capable of recursive updates to parameters (e.g., ion channel density, receptor expression, synaptic weights, and neuromodulatory tone) via Bayesian filtering or on-line Expectation-Maximization, refine structural priors using longitudinal diffusion MRI, and train non-linear components as new data demonstrate biological drift [120,121,122,123]. Ultimately, even model topology may need to be expanded to include additional measurable variables that prove to be relevant to a patient’s dynamics [124].

In addition to integrating biological factors, digital twins must also integrate ecological factors (cognitive baseline, behavioral phenotype, lifestyle, sleep, nutrition, stress, and cognitive load). Twins can simulate how the presence of these ecological factors influences the controlability, metabolic reserve, and glial buffering capacity of an individual and thus optimize interventions to match an individual’s daily-life context rather than laboratory abstractness [125,126,127].

### 3.3. Predictions, Simulations, and Grounded Translational Roles

By far the most significant aspect of a digital twin is its ability to simulate the future trajectories of an individualized dynamical system. Operationally, the twin’s near-term role is to infer transition proximity using critical-slowing, flickering, controllability surrogates, or topological changes, and to project how trajectories evolve within an individualized landscape [128]. By monitoring the individual’s current latent state, the twin can track proximity to the edges of an individual’s basin of attraction and monitor early warning signals (critical slowing down, increasing variance, flickering) indicating that a transition is approaching and thus predict when a transition will occur and whether interventions should be initiated proactively or reactively [115].

In epilepsy, twins aim to predict the gradual increase in the excitability of neurons (extracellular K^+^ balance, inhibition, glial buffering) prior to seizure onset and thus initiate closed-loop stimulation or drug administration prior to the onset of hyper-synchronization [9,129]. In TBI, twins may simulate the restoration of thalamocortical controllability and estimate the window during which stimulation would be most effective to facilitate recovery from disordered consciousness [130]. In addition to predicting the future behavior of an individualized dynamical system, twins can also perform counter-factual experiments. By changing parameters in silico (e.g., kinase/neuromodulator pathways, phase specific stimulation schedules, ECM remodeling, or plasticity rules), twins can estimate the effect of various perturbations on the future trajectory of an individualized dynamical system and estimate the uncertainty associated with each perturbation. This reduces the amount of risk associated with applying a particular therapy empirically [131,132,133].

Additionally, twins can calculate individualized control energy landscapes and thus estimate how much input is necessary to produce a specific transition, identifying “high-leverage” control bottlenecks where small changes in the parameters can lead to large changes in the dynamics of the system [134].

This provides a rational basis for designing therapeutic interventions (stimulation protocols, pharmacologic regimens, future gene- or epigenetic-editing strategies) that are both clinically efficient and physiologically plausible.

Clinical translation is expected to proceed incrementally, through disease-specific examples that are grounded in evidence, rather than through broad statements. Examples of potential near-term clinical applications include personalized seizure forecasting and closed-loop neuromodulation in epilepsy, developing plans for stimulation in disorders of consciousness following TBI, determining optimal rehabilitation intensity/timing after stroke, and detecting the plasticity windows in early neurodegenerative diseases where interventions can potentially expand adaptive repertoires [130,135]. Potential psychiatric applications are beginning to emerge more slowly and include formalizing the distributed neuromodulatory and glial disturbances that bias affective attractors, and evaluating closed-loop or precision-pharmacologic corrections in limited and controlled environments. Potential enhancement applications and co-adaptive brain–computer interfaces are possible longer-term directions; however, they are currently highly speculative and raise many ethical concerns [136].

Digital twins are computational representations of an individual’s brain that are individualized, co-evolving, and dynamically modeled. They assimilate multiple types of data to infer the latent state of the brain, predict how the brain will change over time, and simulate interventions in a probabilistically uncertain framework. Through the combination of data-assimilation techniques and hybrid mechanistic–machine learning architectures, they link molecular and cellular variables to circuit, systems, and behavioral processes. The most well-established and immediate application of digital twins is to predict critical transitions in individualized brain systems, to perform counterfactual tests of treatments, and to optimize closed-loop interventions, whereas the potential for enhancement applications is substantially farther away and subject to strict ethical boundaries [137].

Through this Figure 1, we want to express how digital twins transform neuroscience from passive witnessing, or natural observation, to predictive witness and dynamic steering, representing the computation al basis for precision neurotherapeutics.

### 3.4. Outlook and Future Directions

Some potential digital twin capabilities noted previously are forward-looking as well. The inclusion of these capabilities is in order to provide a notionally possible extension of the current state transition paradigm but does not suggest established clinical validation. Longitudinal monitoring of “slow” variables (immune–glial, metabolic, vascular–glymphatic, or epigenetic markers, for example), may provide better insight into changes in the landscape over time and potentially earlier warning signs of transition risk [138]. However, an end-to-end safety-constrained closed loop system that couples state estimation using digital twin techniques with an adaptive controller is currently purely hypothetical and will need to undergo translational evaluation, identifiability evaluations, and ethical-safety evaluation prior to its use on patients in clinical settings. Additionally, longer-term development-oriented digital twins and co-adaptive brain–computer interface technologies are other potential areas for development but are currently considered to be highly speculative and have a number of limitations in terms of their ethical application [139].

## 4. Adaptive Control Systems: Steering Brain Dynamics in Real Time

Adaptive control allows for closed-loop, real-time trajectory steering in the context of disease as an attractor/trajectory failure, extending digital twin modeling from the reconstruction of the individualized state-space to the closed-loop trajectory-steering. The predictive models or mappings cannot independently alter the underlying dynamics; therefore, adaptive control predicts the onset of pathology, maintains fragile regimes, and guides activity towards adaptive attractors utilizing real-time sensory information, online state estimation, predictive modeling, and feedback actuation, which evolve based upon the changing dynamics of the system rather than static thresholds [140,141].

### 4.1. Neural-Dynamic Control Primitives: Controllability, Observability, and State-Aware Feedback

Adaptive control relies on both controllability (the energetic feasibility of moving the brain from its current attractor to a desired regime) and observability (the ability to determine the brain’s latent states from measurements). Controllability and observability are both difficult to achieve in neural systems due to non-linear dynamics, stochasticity, multi-scale interactions, and dense recurrence; key variables are often latent, indirectly measured and subject to internal and external perturbations [142]. Controllability refers to the energy required and the accessibility of regime changes (e.g., seizure → desynchronized physiology; DOC → wakeful attractor) [143]. Linear Gramian rank criteria are insufficient for determining controllability; instead, local linearizations of manifolds, geometry, and energy-based approximations of accessible reachable sub-spaces and minimal-energy paths are used to provide practical estimates of controllability [144,145].

Observability is achieved through continuous inference of latent state, basin proximity, emergence gradients, and early warning signs of critical slowing or flickering from noisy measurement streams [146]. State and uncertainty maps are produced in real time using particle filters, variational Bayes observers, and unscented Kalman filters to continuously update the belief of hidden variables; this process is also dependent on the output of nested estimators that provide feedback policies as the system’s dynamics drift [147,148]. Therefore, digital twins represent the individualized model of state space, while adaptive control determines the safest, most energy efficient direction(s) of the trajectories contained within the individualized state space.

### 4.2. Architectures and Timing of Adaptive Control: MPC, Adaptive Control, Reinforcement Learning, and Energy-Efficient Steering

Controller architectures for neural systems are rapidly shifting from focusing solely on efficacy, towards incorporating safety, interpretability, and biological relevance. Model Predictive Control (MPC) has been demonstrated to be the most empirical approach thus far [149]. MPC utilizes the digital twin as a generative model to simulate possible future scenarios under different inputs, then it solves a receding horizon optimization problem that minimizes the cost associated with the therapy (reduce pathological oscillations, minimize energy usage, maximize physiological variability) under biological constraints (maximize dose ceiling, timing window constraints, regional specific safety limits) [150,151], allowing for the scheduling of therapies based upon the predicted trajectories of the system, rather than fixed protocols. Due to plasticity, metabolism, and topology drifting, adaptive controllers recursively identify new dynamics and adjust their parameters to maintain stability and performance [152] including model reference and gain scheduled designs [153,154].

If there are no explicit dynamics, reinforcement learning (RL) learns nonlinear policies that guide behavior from reward-structured experience (actor–critic, proximal policy optimization) [155]. However, prior to being utilized clinically, RL must have explicitly defined safety constraints, conservatively explore the solution space, and provide an interpretable representation of the solution. A hybrid approach that pairs MPC priors and bounds with RL refinement and includes robust control layers to ensure tolerance to disturbances and adaptive observer layers to mitigate uncertainty is proposed [156,157].

The precision with which we can steer the brain’s state is limited by the brain’s control-energy landscape [158].

Transitions that align with the dominant modes of the brain are low-energy and less likely to cause harm; optimal control formulations will define the minimum energy pathways that suppress pathological oscillations and/or divert the trajectory of the brain away from maladaptive basins [159]. Additionally, timing is important since many neural processes occur rhythmically; stimulation that is phase specific can significantly enhance the efficiency of modulating the activity of the brain far more effectively than phase random inputs; therefore, adaptive controllers must estimate the oscillatory phase of the brain’s activity on-line and time their stimulation to effect significant transitions with minimal input [160].

Coordination across multiple timescales further enhances the consolidation of changes: rapid (ms) stimulation can initiate a transition, whereas slower interventions (e.g., epigenetic modulation, structural plasticity, metabolic stabilization) can stabilize the transition by modifying the geometry of the long-horizon landscape [161].

### 4.3. Closed-Loop Translation and Forward Limits: Clinical Applications, Multimodal Actuation, and Ethical Constraints

Closed-loop neuromodulation represents the most advanced clinical implementation of adaptive control, which implements a real-time sense–compute–act cycle: measure activity (electrodes, optical sensors, biosensors), predict latent state and trajectories, and actuate via stimulation or timed pharmacology [141,162].

While there exist several clinical implementations of closed-loop neuromodulation (e.g., preictal tracking and timely stimulation of epilepsy patients, beta guided adaptive DBS for movement disorders, investigational thalamocortical stimulation for DOC), they reflect a paradigmatic shift from suppressing reactions to proactively intervening at early bifurcation signatures [163]. Potential future implementations of closed-loop neuromodulation could extend actuation beyond electrical inputs by integrating optical, magnetic, pharmacological, and genetic control surfaces to allow for concurrent modulation of excitability, plasticity, homeostasis, and molecular signaling; phase-locked multimodal control may yield more durable and specific corrections of trajectories [164,165].

As the development of biomarkers and safety margins improves, translation of closed-loop neuromodulation is proceeding in epilepsy, movement disorders, TBI, and stroke rehabilitation, with cautious extensions into affective disorders, OCD, and neurodegeneration [166,167,168]. Long-term molecular feedback control that senses slow variables (transcriptional, immune, metabolic, glymphatic state) and triggers matched molecular actuators remains speculative and requires substantial validation [169].

Limitations to the translation of adaptive control include real-time high dimensional processing, interpretability for clinical trust and regulatory compliance, and ethical issues related to privacy, accountability, equity, and the prevention of unregulated identity/cognition alteration [170]; together these limit the acceptable operation conditions of adaptive control [171].

Adaptive control enables the operationalization of the Section 2 state transition framework by combining observability-based state estimation, controllability-aware energy minimization, and timing-sensitive actuation to direct the individualized trajectories away from pathological basins and towards adaptive attractors. Closed-loop neuromodulation has provided clinical proof of concept, while multimodal and molecularly integrated controller architectures are developing, collectively providing the potential for safe bounded design of transitions of brain states [172].

### 4.4. Perspectives and Future Directions

The many control approaches illustrated previously are speculative in nature and have been provided to provide a basis for potential expansions of the dynamical model; they do not reflect existing clinical validation. Future closed-loop neurostimulation systems may leverage several types of non-electrical control surfaces including optical, magnetic, pharmacological, and genetic to better modulate both fast trajectory modulation and long-term basin remodeling [173]. A possible architecture for long-horizon neurostimulation is molecular feedback control loops where slow variables are sensed (transcriptional, immune, metabolic, glymphatic states) and matched molecular actuators (epigenetic regulation, gene expression control) are activated. However, these designs are hypothetical at this time and will need rigorous validation in terms of safety, identifiability, and translation prior to clinical implementation [174].

## 5. Molecular and Genetic Levers for Landscape Reconfiguration

The brain’s dynamic landscape is governed by its molecular and genetic systems and is capable of reorganization to allow the brain to recover from neurological injury and disease. This reorganization occurs as follows: the adaptive attractors are deepened, the size of the pathological basins is reduced, and the transition routes are opened. Thus, the recent advancements in molecular neurobiology, genome engineering, and systems neuroscience have created the possibility to create plausible methods of dynamically reorganizing the substrate on which cognition, behavior, and disease occur [175]. Below, we describe the primary mechanistic classes of molecular/genetic control and how they are combined with digital twin models and adaptive control to create programmable neurotherapeutics.

### 5.1. Genome, Transcription, and Epigenetic Engineering: Changing the Rules of Plasticity and Excitability

Changing the long-term dynamics of the brain is probably the most direct way to permanently alter the long-term dynamics of the brain is by changing the genomic and epigenomic systems from which excitability, plasticity, and the organization of the networks emerges. The emergence of CRISPR–Cas systems that enable the highly specific editing of the genome and epigenome has created platforms for programmable editing of the genes and regulatory elements. Moreover, CRISPR transcriptional editors (CRISPRa/CRISPRi) connect inactive Cas proteins with activator or repressor domains to provide temporally controlled and potentially reversible regulation of endogenous gene expression [176,177,178].

From a dynamical point of view, this type of intervention can change the slow variables that determine the geometric properties of attractors, i.e., modify the expression of ion channel genes to increase or decrease the excitability threshold, thereby modifying the curvature of the basin and the amount of energy required for transitions to pathological oscillatory regimes; edit adhesion or scaffolding programs to reorganize circuit connections and thereby change the location of the center of mass of attractors and stabilize new regimes [179]. Epigenetic mechanisms constitute a stable and persistent control surface. DNA methylation, histone modification, and chromatin topology regulate the access of the genome to transcription and encode a molecular memory of past activity, environment, and metabolic status and therefore limit future transitions [180,181].

Deactivated Cas proteins that are connected to epigenetic effectors (DNA methyltransferases/demethylases, histone acetyltransferases/deacetylases) are currently being utilized for locus-specific editing of the epigenome [182]. Targeted demethylation or acetylation at the promoter of neurotrophic or plasticity related genes can recreate the window of opportunity for critical periods in adults and decrease the barriers between attractors and make possible the transitions that were previously inaccessible; in contrast, targeted inhibition of drivers of hyper-excitability can deepen adaptive basins and decrease the risk of seizures [183]. Additionally, manipulation of regulators of metaplasticity (CREB, MECP2, HDAC family) can modify the thresholds for synaptic modification, modify the probabilities of transition, and increase the number of states that can be accessed [184]. Moreover, activity coupled epigenetic actuators that respond only to specific conditions of Ca^+^, transmitters, and metabolism provide a methodically accurate manner to conditionally reshape the landscape, as opposed to randomly rewiring it [185].

Neural dynamics are driven by more than neurons. Glial (astrocytes, microglia, and oligodendrocytes) and/or immune signals relay foundational creators of the dynamical landscape of the brain. They generate and prune synapses, provide homeostasis of ions and neurotransmitters, set network excitability, and metabolically support the brain. In epilepsy, neurodegeneration, and psychiatry, there is an increasing awareness that glial and immune signaling could provide pathological attractors. Engaging these mechanisms provides a strongly novel way to sculpt state space [186,187]. Astrocytes, which have a hugely expansive surface area contacting synapses and vasculature, modulate the potassium, glutamate, and lactate concentrations, which are the basic ingredients of network excitability and oscillatory dynamics. If we conceptualize utilizing astrocytic signaling pathways including connexin signaling, aquaporins, and calcium-dependent gliotransmission, we could cause a dramatic shift in the excitability terrain of the system and likely shift the stability of certain attractors. For instance, AQP4 polarization causes a change in glymphatic clearance and interstitial fluid flow that has downstream effects on extracellular ion homeostasis and therefore on neuronal firing [188].

The microglia (the brain’s resident immune cell) likewise alter the synaptic architecture with phagocytosis and from proinflammatory and anti-inflammatory cytokines. The state of microglial activation determines whether to increase the formation and/or elimination of synapses, which influences neuronal excitability and also determines whether to release proinflammatory or anti-inflammatory molecules [189]. And, microglial manipulation of signaling pathways (e.g., CSF1R, TREM2, or NF-κB) will re-configure the topology of the networks and reconfigure the limits of the attractive basins. Activating microglia accelerates pruning adaptations while being neuroinflammatory neuroprotective from sustained adaptations of excitability and synchronous activity that have features of autism [190].

And, oligodendrocytes and myelination dynamics achieve another layer of regulation. The apposition length and the thickness of myelination over the lengths of internode go back to conduction velocity and timing of spikes and synchrony; however, ultimately, myelination circuits the tested landscapes and their regimes. Gene editing or drug manipulation of myelin from Myrf or FGF or others ascertain conduction features that along with distinct activation from neurons and synapses will reshape the tested topology landscape or reshape the landscape itself [191].

### 5.2. Glial–Immune and Metabolic–Vascular Control: Controlling Homeostatic Constraints and Basin Stability

Topology of the landscape geometry is determined not only by neurons, but by glial, immune, metabolic, and vascular functions that establish boundaries on excitability and plasticity. Astrocytes regulate the concentration of extracellular K^+^, glutamate clearance, and lactate supply—all of which are important determinants of the stability of networks and the transition thresholds [192,193]. Gliotransmitter release from astrocytes can modulate the stability and resilience of attractors through connexins, AQP4, Kir4.1 buffering, and Ca^+^-dependent gliotransmitter signaling. For example, AQP4 polarization regulates the clearance of glymphatic interstitial fluid and the flow of interstitial fluids, thereby regulating the stability of spike trains and the structure of the basin [194]. Microglia influence the organization of synapses through activity-dependent pruning and cytokine signaling and therefore regulate the gain and noise characteristics of circuits and attractor stability. Activation state of microglia determines whether the synapses are stabilized or eliminated and whether pro- or anti-inflammatory programs are expressed [189]. Targeting of pathways such as CSF1R, TREM2, or NF-κB can adjust microglial pruning and inflammatory tone and therefore indirectly affect the basin topology in disorders where the pathophysiology involves the movement of trajectories toward maladaptive regimes. Oligodendrocytes provide a different type of control surface: adaptively regulated myelination changes the speed of conduction and the timing of spikes and therefore the synchrony and stability of recurrent loops. Regulation of oligodendrocyte regulators (MYRF, FGF-linked programs) can modify the timing relationships that define large-scale eigenmodes and, after some time, the geometry of attractors [190,191].

Additionally, metabolic and vascular processes limit the energetic requirements for maintaining the attractors. The functions of mitochondria, oxygen delivery, neurovascular coupling, and availability of substrates (glucose, lactate) limit the depth of the basin and the stability of trajectories; failure of these resources can lead to pathological transitions, such as metabolic crisis induced seizures or spreading depressions [192]. Molecular control of regulators of bioenergetics (PGC-1α, SIRT1, AMPK) can improve mitochondrial function and protect against energetic bifurcations and therefore indirectly stabilize adaptive basins [185]. At the network level, interventions that promote neurovascular coupling or support angiogenesis (VEGF- and NO-linked pathways) can expand the range of operation in which the system operates and reduce transition variability during stressful events [195].

Bioelectronic and optogenetic tools can theoretically target metabolic and vascular constraints directly—utilizing closed loop control of blood flow or metabolic enzymes that are activated by neuronal activity to shape the energetic substrate that allows synaptic and genomic plasticity [196,197].

### 5.3. Combining Molecular Levers with Digital Twins and Adaptive Control: Toward Programmable Neurotherapeutics

Ultimately, the largest leverage will be achieved when molecular control is incorporated into closed-loop dynamical frameworks. Digital twins can forecast how genome edits, epigenetic alterations, glial modulation, or metabolic interventions deform the geometry of attractors over weeks to months; adaptive controllers can then time and dose these actuators based on real-time estimates of the state, creating multi-level feedback loops that span millisecond to long term reconfiguration [198].

Synthetic gene circuits represent an especially appealing internal actuator layer: for example, circuits that can detect elevated levels of inflammatory cytokines can activate anti-inflammatory transcriptional programs when attractor stability decreases; circuits that can monitor aberrant oscillatory patterns can stimulate homeostatic excitability countermeasures [199,200].

Therefore, programmable neurotherapeutics represent the likely achievable goal in the near term: interventions do not simply suppress symptoms but progressively reorganize the landscape parameters to restore adaptive repertoires, open plasticity windows, and decrease the likelihood of future pathological transitions [201,202]. Long-term prospects—such as engineering completely novel cognitive attractors or enhancing cognition beyond typical physiological limits—should be viewed as futuristic and speculative and require significant additional empirical evidence and ethical constraints prior to being clinically viable [203,204]. Figure 2 displays the convergence of the molecular and genetic control surfaces (genome/transcription engineering, epigenetic remodeling, glial–immune modulation, metabolic–vascular regulation) and illustrates how their integration with adaptive control can, in principle, deepen adaptive basins, shrink pathological ones, and expand the state space of the brain that is accessible.

### 5.4. Future Perspectives and Directions

The several molecular-control approaches discussed above represent a number of potential future avenues. Rather than implying current clinical validation of the proposed molecular-control approaches, they were described as possible ways in which the transition-based framework can be extended. In theory, activity-coupled epigenetic or transcriptional actuators may provide for the creation of more specific basins through the manipulation of the epigenetic or transcriptomic states of a cell while minimizing the risk of non-specific effects; synthetic gene circuits may enable the construction of internal feedback loops which stabilize the attractor when an early warning structure begins to emerge [205]. However, these types of programmable architectures remain theoretical and will need to undergo extensive translational testing along with safety evaluations over time and scrutiny by regulatory bodies and ethicists prior to being considered for widespread clinical use [206].

## 6. Network Plasticity and Attractor Engineering

Plasticity, as defined in Section 2, as a means to induce changes in attractors by changing the basin shape due to either rapid trajectory steering inputs (Section 4), or slower reshaping of the landscape (Section 5), determines how long the effect of a trajectory change will last, or if it will be maintained as a new attractor basin, new separatrix geometry and new manifold accessability. Therefore, we have divided plasticity into three dimensions for controlling: (1) Synaptic/Metaplastic Rules that modify local basin depths and transition boundaries; (2) E/I Balance that modifies meso-scale basin resilience and transition energies; (3) Oscillatory/Scaffold Structure to modify the time and spatial constraints of the attractor manifold. If molecular and genetic manipulations modify the substrate upon which neurons compute information, and if adaptive control guides the trajectory of neural ensembles with temporal resolution, then it is plasticity that will modify the attractor landscape. Plasticity will specify the way in which synaptic efficacies are changed, the way in which circuits and rhythmic patterns are reorganized, and the emergence, stabilization, and dissolution of different dynamical regimes. More importantly, plasticity is not just the mechanism of learning and memory formation. It is the engine that drives the reorganization and creation of attractors, allowing adaptive basins to be progressively deepened, pathological sinks to be flattened, and previously occluded transition paths to be reopened [207,208,209,210]. We have described below the multi-scale mechanisms of plasticity in terms of three interconnected control planes—synaptic and metaplastic rule sets, excitatory/inhibitory balance, and oscillatory/structural organization—that enable directed attractor engineering.

### 6.1. Synaptic and Metaplastic Mechanisms: The Deepening, Widening, and Shifting of Basins and Transition Rules

At the molecular core of attractor landscape modification lies synaptic plasticity—the activity-dependent strengthening or weakening of neural connections. As mentioned earlier, classical forms of long-term potentiation (LTP) and long-term depression (LTD) continue to represent the foundational mechanisms of synaptic plasticity. Calcium influx through N-methyl-D-aspartate (NMDA) receptors activates a cascade of downstream events, including CaMKII, PKC, and PKA, leading to the phosphorylation and synaptic incorporation of alpha-amino-3-hydroxy-5-methyl-4-isoxazolepropionic acid (AMPA) receptors, thus enhancing synaptic efficacy [211]. Conversely, LTD balances LTP through phosphatase-mediated removal of AMPA receptors and reorganization of cytoskeletal anchoring, thereby weakening synaptic efficacy. Collectively, these two opposing state-changes modify network energy landscapes: Potentiated recurrent loops deepen basins and stabilize attractors, while reduced or pruned loops flatten basins and lower the barrier to transitions into alternative regimes [212].

Synaptic plasticity extends beyond the regulation of synaptic gain. Dendritic translation that occurs locally within dendrites, under the regulatory influence of mammalian target of rapamycin (mTOR) signaling, enables individual synapses to modify their protein composition based on input statistics. Retrograde endocannabinoid signaling similarly fine-tunes the presynaptic release probability, providing an additional feedback layer to the regulation of synaptic efficacy and the stability of attractors [213]. Changes in intrinsic excitability plasticity—through alterations in the expression or phosphorylation of ion channels—also modify the input/output characteristics of neurons, thereby modifying basin stability independent of changes in the wiring diagram [214].

Activity-induced structural plasticity refers to the modification of synaptic morphology through the activity-dependent growth, shrinkage, creation, and elimination of spines. This process is mediated by activity-dependent remodeling of actin filaments, Rho-GTPase signaling, and BDNF-TrkB signaling. Over time, this structural plasticity will modify the network topology, thereby modifying the position of attractor centers and the location of separatrices between them. Furthermore, the activity-driven axonal sprouting and synaptogenesis that occur in response to neural activity will generate novel circuit motifs, thereby enabling truly novel dynamical regimes [215].

Importantly, the threshold for synaptic plasticity is itself plastic (i.e., metaplasticity). Prior neural activity, the state of the neuromodulatory system, and the epigenetic context of the neuron will all modulate the induction rules for LTP/LTD. For example, the phosphorylation of CREB, the histone acetylation, and the remodeling of chromatin will all modulate whether networks are poised for change, thereby determining when attractor remodeling will be possible and in what direction the geometry of the basin will change [216].

Contemporary molecular tools now exist that permit directed manipulation of these rules: Optogenetics can be used to direct spike-timing-dependent plasticity (STDP) at the level of milliseconds, chemogenetics can be used to selectively activate cAMP/PKA or mTOR pathways to promote potentiation or depression, and activity-gated transcription factors can be used to selectively increase the expression of genes involved in plasticity in a state-dependent manner. These technologies can be used to convert plasticity into a directed lever for modifying the directionality of synaptic plasticity and the geometry of basins and transition likelihoods with high specificity [217].

### 6.2. Balance Between Excitatory and Inhibitory Neural Populations as a Stabilizer of Landscape Geometry and Transition Probability

The geometry of attractors is determined by both the strength of synaptic connections and the global balance between excitatory and inhibitory (E/I) populations. Global E/I balance establishes the overall gain of the network, its ability to tolerate noise, and its propensity to transition between different dynamical regimes. An excess of excitatory drive will narrow the funnel-shaped region surrounding a hyper-synchronous pathological attractor, whereas an excess of inhibitory drive will fragment the trajectory of the state-space and reduce the degree of flexibility in the use of cognitive strategies [58].

Different classes of inhibitory interneurons will modify the attractor landscape in distinct manners. For example, parvalbumin (PV) interneurons will synchronize the activity of excitatory populations and stabilize attractors corresponding to the gamma regime. Somatostatin (SST) interneurons will regulate the integration of dendritic potentials and thereby determine which inputs will lead to state transitions. VIP-disinhibitory circuits will regulate plasticity and may either stabilize or destabilize attractors depending on the specific conditions [218]. The efficacy of inhibitory transmission will depend on the maintenance of chloride homeostasis. Transporters such as KCC2 and NKCC1 will establish the reversal potential of GABAA currents; changes in these transporters can result in decreased efficacy of inhibitory transmission, and even in the case of some types of neurons, the conversion of inhibitory transmission to excitatory transmission. The latter will dramatically alter the curvature of basins and the location of separatrices separating attractors. Neuromodulatory substances will also establish the magnitude of inhibitory tone; acetylcholine will decrease inhibitory tone and enhance plasticity during learning, norepinephrine will sharpen the temporal precision of neural responses, and serotonin-containing systems will modulate the gain of the network and the depth of affective attractors through receptor-specific interactions [219,220]. The implications of these mechanisms are that there are specific routes for intervention. For example, PV-interneuron targeted optogenetic stimulation can suppress pathological oscillations, disinhibition of prefrontal circuits may facilitate escape from rigid affective basins, and pharmacologic correction of chloride transporter function can restore normal inhibitory function in the case of epilepsy and certain neurodevelopmental disorders. When coupled to adaptive controllers, these routes can be used to dynamically regulate E/I balance to stabilize trajectories or induce escape into healthier attractors [221].

### 6.3. Oscillatory and Structural Plasticity: Temporal Scaffolding, Topology Change, and the Construction of New Attractors

Oscillations serve as the temporal framework for the organization of the activity of neural populations and provide the basis for the coordination of activity across multiple spatial scales. For example, gamma frequency oscillations support local binding and working memory, theta frequency oscillations support hippocampal–prefrontal communication, and slow oscillations support global transitions such as those occurring during the sleep–wake cycle [222].

Therefore, oscillatory coherence will modify the stability of basins: Increased coherence of gamma frequency oscillations can deepen the working memory attractor, while disruption of theta–gamma frequency coupling can reduce the integrity and flexibility of basins. Pathological beta frequency synchronization in Parkinson’s disease and other movement disorders represents a maladaptive oscillatory attractor that is unresponsive to transitions [223,224]. Entrainment-based stimulation can re-establish flexibility by modifying the oscillatory scaffolds that govern the state-space, thereby increasing the width of the transition corridor. Temporal structure will couple with synaptic plasticity: The synaptic strength of synaptic connections will be modified by the timing of pre-post spikes, and the oscillatory phase will determine whether the timing will favor potentiation or depression of synaptic strength. Therefore, phase-locked stimulation can be used to direct the direction of synaptic plasticity and anchor new attractor structure through temporally aligned inputs [225]. Cross-frequency coupling will add another layer: Slow rhythms will gate windows in which fast oscillations will drive synaptic modification, thereby providing a route for nested time-engineering of attractors. Closed-loop stimulation can be used to estimate the phase of oscillatory activity in real-time and apply perturbations that are less than the period of oscillation; tACS can be used to entrain macroscopic oscillatory rhythms; and optogenetic methods can be used to sculpt rhythms at the level of microcircuits. Collectively, these methods can be used to engineer the geometry of basins—deepen adaptive regimes, compress pathological ones, and bias future trajectories [226].

Structural plasticity will scale the effects of oscillatory plasticity by modifying the topology of the network itself. Spine turnover, axonal sprouting, synaptogenesis, and large-scale white matter remodeling will all modify the connectivity graph that specifies the manifold of attractors [227]. Perineuronal nets and ECM components will limit the extent of adult rewiring; however, enzymatic degradation of these constraints can reopen developmental-type plasticity windows and extend the accessible state space [228].

Activity-dependent myelination, governed by FGF, ErbB, and Notch pathways, will modify the velocity of action potential propagation and synchrony and thereby remap the eigenmodes that support large-scale attractors. Longitudinal imaging and closed-loop training/stimulation can be used to iteratively guide this rewiring and gradually construct the topology of the circuit to stabilize newly engineered dynamical regimes [229].

Therefore, the implication is that attractor engineering can progress from restoration to the construction of entirely new adaptive configurations. Coordinated modulation of synaptic rules, E/I balance, oscillatory scaffolds, and structural topology can eliminate maladaptive basins, stabilize recovery-supporting regimes, and create new task-specific attractors in rehabilitation or disease contexts [230,231].

Claims of enhancement beyond typical physiological limits are forward-looking and should remain explicitly speculative pending robust empirical validation [232,233,234]. Table 2 provides a summary of these plasticity control surfaces and their timeline, indicating where interventions can be used to most plausibly modify the geometry of attractors and the structure of transitions. Plasticity is the biological machinery of attractor engineering. Through the coordinated modulation of synaptic and metaplastic rules, E/I balance, oscillatory synchrony, and structural topology, plasticity can deepen adaptive basins, flatten pathological sinks, and open new transition corridors. The development of modern molecular tools, neuromodulation techniques, and closed-loop control methods have transformed plasticity from a passive property to a directed design surface for recovery-oriented state-space remodeling [235].

### 6.4. Concluding Perspectives and Future Directions

The potential applications of Attractor Engineering continue to be speculative and have been added to illustrate potential expansions on the Plasticity-based Control Planes as opposed to validated in clinical application. Theoretically, increasingly precise (phase locked or activity gated) intervention strategies could provide for more specific reconfiguration of oscillation scaffolding and synaptic rules to support stabilization of recovery basins at less energy expense [246]. Additionally, theoretically, future multi-plane coordinated strategies that focus on both restoration of function and creation of new adaptive functions may serve as a long term direction; however, design(s) of this nature will need to undergo rigorous empirical testing/longitudinal safety evaluation/ethical governance prior to broad clinical applicability [247].

## 7. Translational Horizons: Empirically Grounded Pathways for State-Space Neurotherapeutics

To translate brain disorders based on failures of state-space navigation, translation should begin at a point where (i) state transitions are quantifiable, (ii) controllability has been shown to exist in humans, and (iii) a connection exists between dynamical biomarkers and clinical outcomes. As such, we will primarily focus on two examples with the most empirical footing: epilepsy, and Parkinson’s disease treated with deep brain stimulation (DBS). We will use these examples to illustrate how digital twins, adaptive control, and molecular/plasticity levers can be translated from a conceptual level to a feasible clinical strategy. We will reserve broader applications (other neurodegenerative, traumatic, psychiatric, or enhancement uses) to speculative possibilities that remain forward-looking and will therefore not develop them here.

### 7.1. Epilepsy: Predictive Biomarkers of Bifurcations and Prevention of Transition into Ictal Attractors

Epilepsy is paradigmatically a state-transition disorder: seizures occur when a trajectory crosses a bifurcation into a hyper-synchronized attractor. The regimes leading up to ictus typically have “early warning signs” indicating that the trajectory is approaching the ictal separatrix, i.e., critical slowing down, increasing autocorrelation and variance, and transience in flicker. However, conventional anti-seizure medications generally decrease global excitability and do not utilize these dynamical indicators to guide timely trajectory correction [248,249].

Digital twins of patients can be developed using multimodal data streams (e.g., electrophysiological, structural/functional connectivity, extracellular ionic measures if available, astrocytic polarization including aquaporin-4, microglial inflammatory tone, and transcriptomic context in research environments) to create patient-specific maps of seizure basins and to provide estimates of the distance to bifurcation boundaries in seconds to minutes prior to an ictus [250]. This formalizes how microdomain K^+^ buffering, AQP4 dependent clearance, and GABAergic inhibition interact to determine the size of the basin and the threshold for transitioning to another basin, while identifying pre-ictal shifts (e.g., subtle increases in glutamate concentration or manifold reduction) that could potentially be overlooked by surface readouts [251]. The forecasts provided by digital twins can be used to generate adaptive control solutions. Responsive neurostimulation algorithms have demonstrated that closed loop stimulation can destabilize an ictal basin before it becomes fully recruited. Additionally, phase locked patterned stimulation that is synchronized with the dominant eigenmode(s) and phase of oscillation can reduce the energy required to control stimulation while maximizing its effectiveness. Finally, closed loop pharmacology provides a complementary actuator that can provide focal inhibitory boosts or excitability dampening at the precise time when the likelihood of crossing into the ictal basin is highest [252].

Slow variable landscape modification can alter the likelihood of recurrence over weeks to months. Levers for modifying the slow variables include CRISPR based modifications of the programs of ion channels associated with the pathophysiology of epilepsy (e.g., SCN1A or KCNQ2 in specific subtypes), the restoration of Kir4.1 mediated K^+^ clearance by astrocytes, and the targeted attenuation of pro-convulsive inflammatory cascades (e.g., NF-kB/NLRP3 axis) [253]. From a dynamical perspective, the translational agenda is multi-layered and includes both rapid trajectory correction to prevent entry into the pathological attractor and gradual modification to decrease the size of the seizure basin and restore the ability of trajectories to transition between attractors [254].

### 7.2. Parkinson’s Disease: Adaptive Deep Brain Stimulation as Steerable Control of Trajectories in Cortico-Basal Ganglia Attractors

Parkinson’s disease demonstrates the best current empirical evidence that neural trajectories can be steered in humans via closed-loop control. The pathological synchronization of beta frequency in cortico-basal ganglia (BG) loops represents a maladaptive oscillatory attractor that constrains motor repertoire and prevents transition between different states. Although conventional DBS improves symptoms, open-loop stimulation does not respond to changes in the state of the system at a given instant [255]. There are already clinical demonstrations of the feasibility of state-aware control in adaptive DBS systems. By measuring beta band activity in real-time and modulating the amplitude, timing, or pattern of stimulation according to the measured activity, these systems reduce pathological synchronization, preserve physiological variability, and decrease side effects. Within the dynamical framework, beta frequency oscillatory power can serve as an observable surrogate for distance into the pathological basin; stimulation provides a controllable input that pushes the trajectory toward more flexible beta-gamma regimes that support adaptive motor states [256].

Digital twins can expand upon this by creating personalized versions of the control problem. Using patient-specific BG-cortical models that incorporate structural connectivity, stimulation response curves, and longitudinal clinical dynamics, digital twins can identify those nodes, frequencies, and phases that represent low-energy control bottlenecks. Adaptive controllers can then calculate eigenmode aligned trajectories that minimize the energy required for control while maximize the restoration of motor attractors [257].

Neuromodulatory or epigenetic strategies that protect synaptic function, maintain glial homeostasis, or support metabolic needs can stabilize the landscape that DBS is navigating through, but the empirically supported core of the translational agenda remains the validated closed-loop DBS platform. Therefore, Parkinson’s disease is a real-world link from dynamical theory to clinical trajectory engineering [258].

## 8. Conclusions: Toward Programmable Neurodynamics

Neuroscience is at the beginning of a profound transition. For over one hundred years, the field has achieved significant success in mapping anatomical structures, identifying neural signal patterns, and cataloging molecular mechanisms underlying brain function. While this research has demonstrated an immense degree of complexity in the organization and functioning of the brain, many clinical treatments remain primitive and largely symptomatic. They seek to suppress symptoms instead of addressing the dynamic processes that create them.

The advances described below suggest a different future for neuroscience: one where the brain is both modeled and controlled to achieve specific states and functions, and across multiple levels of biological organization.

Conceptually, the largest change is how we understand the brain. Rather than viewing the brain as a relatively fixed anatomical entity, the brain should be viewed as a high-dimensional dynamic system in which the generation and failure of function occurs as trajectories within a given attractor landscape. Within this framework, the signaling networks composed of molecules determine the properties of excitability and communication between neurons and glia; plastic changes in synapses and structure modify the connectivity graph that constrains the dynamics of the brain; oscillations and synchronization regulate the transitions between states; and behavior represents the path the brain takes through state space. Therefore, disease represents a distortion of the geometric features of the landscape of brain function—collapse of adaptive attractors, expansion of pathological basins, and loss of transition channels that support cognition and behavior.

There are several converging tools that make possible the practical application of dynamical interventions. Digital twins (1) combine molecular, electrical, structural, metabolic, and behavioral data into adaptive computational representations of the brain that estimate the current brain state, predict the likely trajectory of future brain states and evaluate potential interventions in silico. Adaptive control systems can use these predictions to apply time-dependent and state-specific perturbations that stabilize potentially unstable regimes and/or direct the trajectory of the brain away from pathological “sinks” (2). There are molecular and genetic techniques available to adjust the rate of evolution of slower variables—such as adjusting gene expression profiles, editing or sculpting epigenetic memory, modifying the signaling between glia and the immune system, and replacing/repairing metabolic or vascular constraints (3). Attractor engineering using plasticity provides a means of actively controlling the process of reorganizing the brain, thereby providing an active control surface to modify the geometric characteristics of the basins of attraction, restore previously lost transition paths, and even construct entirely novel dynamical regimes. Collectively, these areas represent the beginnings of an emerging field of programmable neurodynamics—the ability to sense, model, and control the brain’s multi-scale state transitions in order to alter the rules governing those transitions, rather than simply reacting to their effects. Within a well-defined translational scope, this agenda is tangible today. Seizures can be predicted and stopped prior to the completion of ictal bifurcations. Deep brain stimulation using adaptive DBS has provided real-world evidence that pathological oscillatory attractors can be destabilized, and trajectories directed toward physiological states in patients with Parkinson’s disease. On larger timescales, molecular and plasticity-based interventions may be able to re-expand the repertoire of behaviors in individuals with neurodegenerative diseases and rebuild the postinjury transition architecture in disorders of consciousness. Although these objectives are difficult, they are theoretically grounded in a dynamical view of disease and recovery.

However, achieving this vision will require more than the development of technologies. It will require more fundamental theoretical links between the molecular, cellular, circuit, and behavioral dynamics of the brain that can be translated into coherent causal models. It will require thoughtful and effective ethical and regulatory frameworks for interventions that have the potential to alter cognition, mood and identity. Most importantly, it will require humility: the brain is complex and will forever exceed our ability to anticipate fully; and thus, dynamical engineering should remain focused on alleviating suffering, improving function, and enhancing human flourishing, rather than seeking to exert control for its own sake. The work ahead is significant, however, the direction is becoming increasingly clear. As digital twins improve in fidelity, adaptive controllers improve in reliability, molecular interventions improve in precision, and attractor engineering improves in intentionality, neuroscience can move from description to prediction to active manipulation of brain dynamics. In this progression, the key clinical question shifts from “How does the brain operate?” to “How can we help it to operate in a way that is stable, flexible, and resilient?”.

Similar to how the twentieth century mapped the structure of the brain, the twenty-first century may learn to guide the dynamics of the brain. This is not simply a modification of approach, but a redefinition of disease, recovery, and treatment—one that can significantly redefine the practice of clinical neurology and psychiatry and deepen our understanding of what constitutes the self. The transition has already started, and its most important chapters will be authored by sustained collaborative research between dynamic systems theorists, neurobiologists, AI researchers, and clinically oriented researchers at the interface of these fields.

## Figures and Tables

**Figure 1 ijms-27-00122-f001:**
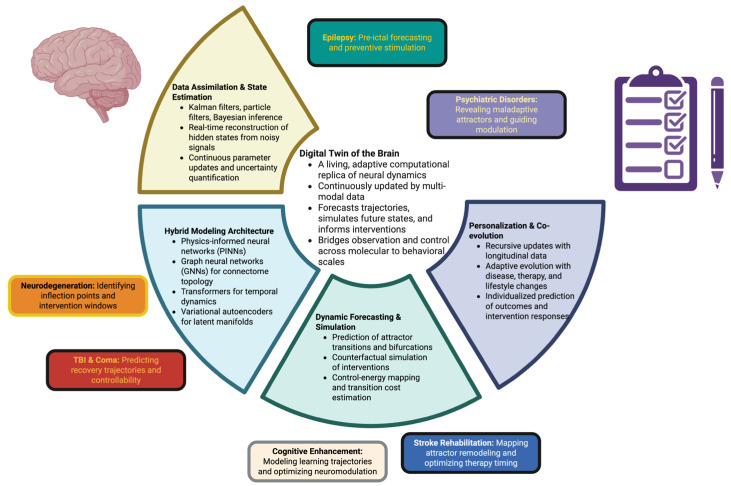
Digital twins, multiscale computational mirrors of brain dynamics and structure. Digital twins are dynamic, self-adaptive computational models, which are data-driven, simulate latent states of the brain, predict future behaviors, and test hypotheses in a virtual environment. They model molecular, cellular, circuit, and behavioral-level data together with their interactions; therefore, digital twins link micro-level mechanisms to cognitive functions and pathological changes; further, they enable the use of counter-factual simulations and develop tailored predictive therapies.

**Figure 2 ijms-27-00122-f002:**
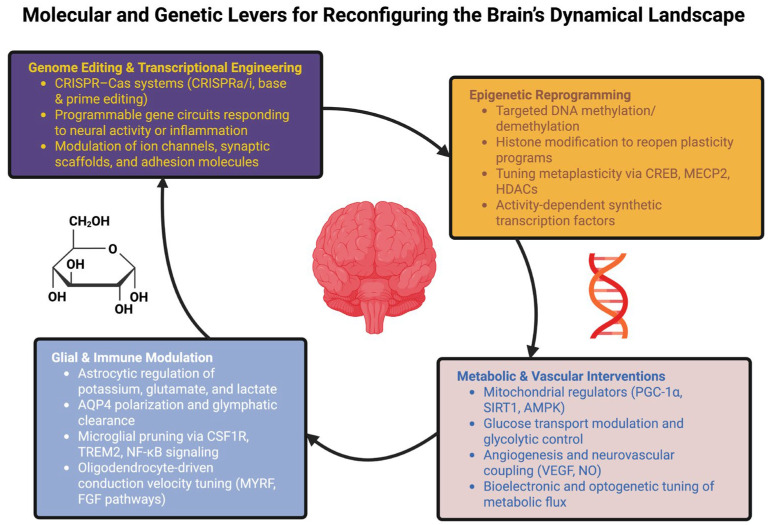
Molecular and genetic interventions as fundamental levers for reconfiguring the brain’s dynamical landscape. CRISPR-based genome/transcriptional editing, epigenetic reprogramming, glial–immune modulation, and vascular–metabolic regulation affect slow dynamics that modify the topological features of the attractors. These manipulations may increase depth in an adaptive basin, decrease width in a pathological sink, prolong plasticity windows, or provide the molecular basis for programmable neuro-therapeutic applications in closed loop-dynamical systems.

**Table 2 ijms-27-00122-t002:** Mechanisms and emerging interventions for network plasticity and attractor engineering.

Domain	Core Mechanisms	Dysregulation	Therapeutic/Engineering Opportunities	References
Synaptic plasticity	NMDA-Ca^2+^ signaling drives LTP/LTD (CaMKII/PKC/PKA → AMPAR insertion/removal). Local dendritic translation (mTOR). Retrograde endocannabinoids tune presynaptic release. Intrinsic excitability plasticity via ion-channel regulation.	Weak LTP/LTD → shallow adaptive basins, unstable memory attractors. Excess potentiation → deep pathological basins (epilepsy). Faulty trafficking/translation → faster decline.	Optogenetic STDP. Chemogenetic steering of cAMP/PKA or mTOR. Activity-gated TFs for plasticity genes. Metaplasticity tuning (CREB, chromatin).	[88,236]
Excitation–inhibition balance	PV+ interneurons synchronize gamma states. SST+ gates dendrites. VIP+ enables disinhibition. KCC2/NKCC1 set GABA polarity. ACh/NE/5-HT shift inhibitory tone and basin depth.	Too much excitation → hypersynchrony, seizure basins. Too much inhibition → fragmented trajectories, low flexibility. KCC2 loss → GABA becomes depolarizing.	PV-optogenetic suppression of hypersynchrony. VIP-disinhibition to exit rigid mood basins. KCC2/NKCC1 pharmacology. Closed-loop E/I tuning.	[237,238]
Oscillatory dynamics	Gamma supports binding/WM. Theta supports HPC-PFC coordination. Cross-frequency coupling gates plasticity. Phase controls STDP direction.	Beta trapping in PD. Loss of theta–gamma coupling → unstable memory states. Low coherence → reduced repertoire.	Phase-locked closed-loop stimulation. tACS entrainment. Sub-cycle optogenetic phase control. Cross-frequency restoration.	[223,239]
Structural plasticity	Spine turnover (actin; Rac1/Cdc42/cofilin). Axonal sprouting/synaptogenesis (BDNF, semaphorins). Perineuronal nets set adult limits. Activity-dependent myelination (FGF/ErbB/Notch) tunes timing and eigenmodes.	Low spine turnover (aging/AD). Rigid ECM blocks rewiring. Demyelination disrupts synchrony.	Perineuronal-net digestion to reopen windows. Molecular biasing of synaptogenesis. Closed-loop stimulation/rehab to stabilize new paths. OPC activation + training for remyelination.	[240,241]
Engineered attractors	Joint tuning of synapses, E/I, rhythms, topology. Higher dimensionality → flexibility. Deeper selected basins → stable attention/memory/mood.	Fewer attractors → limited recovery. Shallow basins → fragile dynamics. Rigid landscapes → poor rehab.	Strengthen recurrent loops post-stroke. HPC-PFC rewiring for new memory strategies. Rhythm/neuromodulator shaping for stabilization or enhancement. Digital twin-guided multiscale design.	[242,243,244,245]

Plasticity is the biological machinery that enables deliberate reshaping of the brain’s dynamical landscape. Across synaptic, inhibitory, oscillatory, structural, and integrative levels, plasticity determines how attractor basins are formed, deepened, dissolved, or newly constructed. Dysregulation at any level destabilizes network dynamics and narrows the brain’s adaptive repertoire, but new strategies—from optogenetic STDP induction, chloride homeostasis modulation, and phase-locked stimulation to enzymatic ECM remodeling, digital twin-guided rewiring, and multi-scale attractor construction—are transforming plasticity into a programmable design tool. These approaches promise not only restoration of lost functions but the augmentation of cognitive capacity and resilience.

## Data Availability

The data presented in this study are available on request from the corresponding author.

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
