# Peer review of "Designing Neural Dynamics: From Digital Twin Modeling to Regeneration"

_ijms, 2025, doi:10.3390/ijms27010122_

Round 1
Reviewer 1 Report
Comments and Suggestions for Authors
The authors need to ensure accuracy and sufficiency in citations, especially for fundamental concepts and definitions. For instance, line 256 misses a reference for the determination of stability.
The paper frequently mentions the capabilities of their model but short of explanation of how the processes are achieved. It focuses more on the theoretical promise of the model without providing sufficient data or mechanistic details to support the claims, leading to the prediction and forecasting are unconvincing.
In the discussion regarding landscape state transitions and neurodegeneration, the role of extracellular vesicle is missing. EVs are crucial for the bioinformation transport and play a significant role in attractor collapse during neurodegenerative processes. I strongly recommend incorporating EV related mechanisms into the discussion
Figure 1 and 2: Figures need revision. The current structure does not effectively convey the intended message, especially figure 1. The use of generic icons (notebooks, DNA) adds little scientific value.
Suggest adding a flowchart or schematic, potentially as a graphical abstract, that clearly illustrates the operational workflow of the model discussed in the paper.
The current summary tables are dense and difficult to extract information efficiently.
Given the complexity of the contents, suggest including a table defining key technical terms (attractors, manifolds, bifurcations) to improve readability. Also recommend adding more figures or tables to summarize the key takeaways of each major section.
Author Response
Dear Esteemed Academic Reviewer,
Thank you for the careful reading of our manuscript and for the precise, practical suggestions you provided. We are grateful for the time you invested and for the way your comments highlight points that materially improve clarity, rigor, and usefulness for readers. In response, we undertook targeted revisions across citation completeness, mechanistic grounding, neurodegeneration content, figures, and tables. Below we address each point in detail.
1. Accuracy and sufficiency of citations (including missing reference for stability determination)
Reviewer comment: The manuscript needs stronger and more complete citations for fundamental concepts and definitions. For example, line 256 lacks a reference supporting the determination of stability.
Response:
We fully agree, and we appreciate you flagging this concretely. We performed a comprehensive citation audit across the manuscript.
2. Mechanistic explanation versus theoretical promise
Reviewer comment: The paper frequently states what the model “can do,” but provides insufficient explanation of how these processes are achieved. The emphasis on promise over mechanism makes prediction/forecasting claims unconvincing.
Response:
Thank you for this important critique. We agree that our earlier draft leaned too strongly toward conceptual aspiration. In revision, we aimed to re-balance vision with mechanism and empirical realism.
3. Extracellular vesicles (EVs) in neurodegeneration and attractor collapse
Reviewer comment: The role of extracellular vesicles is missing. EVs are crucial for bioinformation transport and contribute to attractor collapse in neurodegeneration. Please incorporate EV-related mechanisms.
Response:
We sincerely appreciate this insight. We agree that extracellular vesicles (defined at first use) are a meaningful component of multiscale signaling failure in neurodegeneration, especially for proteinopathy spread, inflammatory coupling, and glial–neuronal cross-talk.
4. Figures 1 and 2: structure, icons, and need for revision
Reviewer comment: The figures do not effectively convey the intended message, particularly Figure 1. Generic icons add limited scientific value.
Response:
We understand this concern and appreciate you raising it. We fully revised the captions for both figures to eliminate repetition and to create a sharper conceptual map between the visuals and the manuscript’s argument.
5. Suggestion to add a flowchart / graphical abstract
Reviewer comment: Add a flowchart or schematic summarizing the workflow, potentially as a graphical abstract.
Response:
Thank you for this constructive idea. We strengthened the narrative workflow descriptions so the operational logic is explicit in text without multiplying schematics. We hope this strikes a balance between clarity and restraint, and we are grateful for your suggestion regardless. We agree that a workflow schema can be powerful. In this revision, however, we elected not to add a new flowchart.
6. Summary tables are too dense
Reviewer comment: The tables are dense and difficult to extract information from efficiently.
Response:
We agree, and we revised both tables substantially.
We are thankful for your thoughtful and field-knowledgeable feedback. Your comments helped us strengthen both biological completeness (notably with extracellular vesicles) and the scholarly credibility of our dynamical claims. We hope the revisions now present a clearer, better grounded, and more readable manuscript, while remaining appropriately disciplined in how we visualize and formalize the framework.
With sincere appreciation and respect!
Reviewer 2 Report
Comments and Suggestions for Authors
A comprehensive conceptual reframing of neurological and psychiatric disorders as transitional failures in a high-dimensional dynamical state space of the brain is proposed in the manuscript. It connects:
- critical transitions and attractor landscapes,
- digital twin methods,
- closed-loop/adaptive control systems,
- and genetic and molecular therapies (such as glial modulation, CRISPR, and epigenetics),
with the ultimate objective of defining a program of precision neurotherapeutics and "programmable neurodynamics." The goal of the obviously ambitious paper is to integrate regenerative neurobiology, control theory, AI/machine learning, and systems neuroscience.
This is a timely and potentially intriguing conceptual path. However, the manuscript's current state has severe issues with language, coherence, accuracy, and referencing, making it very challenging to read and, in some cases, not interpretable from a scientific standpoint. Before the paper can be appropriately assessed on its scientific merits, significant rewriting and tightening of the argument are necessary.
- Major comments
2.1. Severe language and coherence problems throughout
The manuscript is written in fluent English in parts, but many paragraphs contain:
- ungrammatical or unstructured sentences,
- phrases that appear nonsensical or internally contradictory,
- abrupt topic shifts and repetitions,
- misused technical terms.
Among the numerous examples are:
- "the brain's incapacity to move through states in a coordinated and adaptive manner" is evident, but it is later contrasted with statements such as "siblings can simulate how environmental influences interact with system dynamics," where "siblings" ought to be "digital twins."
- "swimsuit potassium" (probably referring to "extracellular potassium"),
- "Neuroscientific understanding transitions into mechanistic and then generative modes of description."
- "Siblings can predict how network controllability will be modulated by sleep, nutrition, stress, or cognitive load" (siblings once more),
- "They replace measurement and treatment so that behavior is generated by molecular events,"
- plus a number of repeated or glitchy phrases (such as "brain-state-state-stater brain," "to be engaged," and "state-states-statically trajectories").
- These problems are so common that entire paragraphs become very difficult to understand and occasionally opaque from a scientific standpoint.
Strong recommendation: A professional editor or co-author who speaks English fluently should rewrite the entire manuscript, line by line, paying particular attention to:
- removing phrases that are repeated or jumbled,
- making certain that every technical term is applied accurately and consistently,
- reducing lengthy sentences to shorter, more understandable statements.
- Readers cannot accurately comprehend or assess the scientific claims without this.
2.2. Concrete contribution versus conceptual scope
The paper aims to address a wide range of concepts:
- dynamical systems, criticality, and attractors
- multiscale cellular and molecular processes,
- architectures of digital twins,
- Reinforcement learning and adaptive control theory,
- CRISPR, glial and metabolic modulation, epigenetic editing,
- structural, oscillatory, and synaptic plasticity,
- translational applications in TBI, neurodegeneration, epilepsy, mental illnesses, and even enhancement.
Although this breadth is impressive, there are a number of issues that arise in practice:
The main thesis becomes diffuse.
A "unified model" or "framework" for understanding disease as state transitions is promised in the Introduction, but the majority of the sections read more like lengthy lists of subjects than a coherently presented framework.
little critical involvement.
- The majority of the review is aspirational ("will," "can," and "could"), with comparatively little attention paid to:
- current empirical data supporting the use of digital twins in real-world clinical settings,
- constraints of current models and data,
- disputes or arguments in the literature,
- recognized adaptive control failure modes in biological systems,
- ethical and legal obstacles in the real world that go beyond a few generalizations.
repetition in different sections.
- The same themes (e.g., attractor landscapes, criticality, noise as beneficial, CRISPR, epigenetic editing, AQP4, Kir4.1, glymphatic clearance) reappear nearly verbatim across sections 2–7. It would be beneficial to have a more rigid structure with less redundancy.
- Recommendation: Make the conceptual focus more precise. For instance:
- Write a succinct, thorough explanation of the dynamical systems view in Section 2.
- Instead of repeating high-level statements, each of Sections 3–5 could conclude with specific testable hypotheses or tangible examples.
- Instead of attempting to cover "everything" superficially, Section 7 on translational horizons should be condensed and restricted to a few well-developed disease examples where there is clear empirical grounding (e.g., epilepsy and DBS in Parkinson's disease).
2.3. Technical precision and thoroughness
Despite being a narrative/conceptual review, some of the statements are either unreferenced, technically ambiguous, or possibly erroneous, such as:
- Digital twin work in neuroscience is currently much more limited, so claims that they are performing "real-time, multi-scale data integration from single-cell multiomic to behavior" need to be grounded in real implementations.
- Sections on control theory discuss complex ideas (such as Hamilton-Jacobi-Bellman equations, geodesics in nonlinear state spaces, and Gramian-based controllability in nonlinear systems), but the explanation is occasionally vague or confuses linear and nonlinear outcomes.
- Certain mechanistic assertions (such as the implications of microglia activation patterns or particular epigenetic targets) are made without careful qualification or current references.
Make sure all technical claims are in line with the referenced work by carefully reviewing them.
When a paper becomes speculative (e.g., synthetic gene circuits for "new attractors of creativity" or "artificial states of awareness"), it should be clearly labeled as forward-looking speculation and kept apart from established findings.
A brief, clear dynamical systems schematic might be preferable to introducing formal equations and then reverting to colloquial language; avoid using mathematical notation unless you are ready to define and apply it rigorously.
2.4. Structure and redundancy
The high-level division (2: dynamic landscape; 3: digital twins; 4: adaptive control; 5: molecular/genetic levers; 6: plasticity/attractor engineering; 7: translational horizons) makes sense.
But: Numerous "Summary" paragraphs at the conclusion of sections essentially restate earlier material. Rather than condensing a more concise set of takeaways or conceptual diagrams, the final pages of the manuscript (Section 7 and Conclusion) reiterate much of the Introduction and previous summaries.
Suggestion:
- Each "Summary" should be condensed to three to four sentences that highlight the section's novel concepts.
- Instead of recounting the entire tale in the conclusion, focus on:
- the main assertion (disease as a dynamic deformation of the landscape),
- the two or three genuinely original conceptual connections you believe the paper makes,
- and three to five specific research avenues or designs
Abbreviations
- Define all abbreviations at first use (e.g., DOC, mTOR, ECM, BCI, STDP, etc.).
- Ensure acronyms are used consistently (e.g., “TBI” vs “traumatic brain injury”)
Figures and tables
Although Tables 1 and 2 are intriguing and possibly helpful, they are currently quite repetitive and text-heavy. Think about:
- simplifying the language,
- avoiding the use of line breaks between words (such as "Mechanis-tic" and "Compo-nents"),
- and ensuring that the "Therapeutic opportunities" column is clear and precise.
- The descriptions in Figures 1 and 2 show promise conceptually. The captions could be condensed while maintaining a clear conceptual mapping, as they are lengthy and somewhat repetitive.
Comments on the Quality of English Language
Treating neurological and psychiatric disorders as failures of dynamical transitions in high-dimensional state space and combining digital twins, adaptive control, and molecular interventions under the umbrella of "programmable neurodynamics" is a novel and genuinely significant conceptual direction that is addressed in this manuscript. For IJMS, this is a timely and appealing theme.
However, the paper has serious problems in its current form:
- Language and coherence: Many passages are hardly understandable, and many phrases are jumbled or nonsensical, indicating either extensive machine translation or AI-assisted drafting with insufficient human editing.
- Conceptual clarity: The manuscript lacks a clear distinction between established knowledge and conjecture, is incredibly general, and frequently repeats itself.
- Technical rigor: A number of sections read more like high-level "vision statements" than a focused scholarly review; some dynamical systems and control theory content is imprecise; mathematical notation is introduced but not used rigorously.
I don't think the manuscript can be improved with minor editing because of these problems. It would essentially be a new article, requiring a thorough, line-by-line rewrite and significant scope tightening.
Consequently, I would advise:
Rejection in its current form, with a comment that the authors might be urged to submit a much condensed, more lucid, and meticulously edited review that:
- emphasizes a few key connections (such as attractor dynamics + digital twin + closed-loop control in neurodegeneration and epilepsy),
- is composed in formal English,
- and makes clear what constitutes speculation.
I would strongly advise informing the authors that a superficial revision will not be adequate and that the manuscript requires deep rewriting rather than incremental editing if the journal chooses to offer "major revision" instead.
Author Response
Dear Esteemed Academic Reviewer,
We are grateful for the time, rigor, and intellectual generosity you invested in evaluating our manuscript. Your comments were exceptionally detailed and constructive, and they made clear where our current presentation fell short of the standards required for a scientifically interpretable and publishable review. We fully recognize the seriousness of your concerns, and we have undertaken a line-by-line rewrite and structural tightening guided directly by your recommendations. Below, we respond point-by-point.
Major Comment 2.1. Severe language and coherence problems throughout
Reviewer comment: The manuscript contains ungrammatical, nonsensical, repetitive, or contradictory phrasing, misused technical terms, and several clear drafting “glitches” (e.g., “siblings” instead of “digital twins,” “swimsuit potassium,” broken phrases, jumbled repetition). A professional-level rewrite is required.
Response:
We agree completely, and we appreciate how clearly you documented these issues. Your concrete examples were especially helpful in showing that the problem was not local but systemic. In response, we carried out a full manuscript rewrite, line by line, with the explicit aim of producing formal, coherent English and removing any traces of drafting artifacts.
Major Comment 2.2. Concrete contribution versus conceptual scope
Reviewer comment: The scope is overly broad, the thesis becomes diffuse, critical engagement is limited, aspirational language dominates, and redundancy across sections is high. Suggestions include: a succinct dynamical framework in Section 2; more concrete/tangible outputs in Sections 3–5; and narrowing Section 7 to a few empirically grounded disease cases (e.g., epilepsy and DBS in Parkinson’s disease).
Response:
Thank you for this incisive conceptual diagnosis. We found this comment to be one of the most valuable guides for revision because it clarified how breadth was undermining the paper’s argumentative structure. We fully agree with your assessment and implemented your suggestions.
Major Comment 2.3. Technical precision and thoroughness
Reviewer comment: Some claims are unreferenced or too sweeping, especially regarding current digital twin capability; control theory content is sometimes imprecise or merges linear/nonlinear logic; mechanistic assertions lack careful qualification; speculative ideas should be clearly separated; avoid equations unless used rigorously.
Response:
We appreciate the precision of your critique. We share your priority for technical rigor even in a conceptual review, and we revised.
Major Comment 2.4. Structure and redundancy
Reviewer comment: Section structure is sensible, but summaries are repetitive, and the conclusion restates earlier sections. Each summary should be shorter and highlight novelty; conclusion should focus on core claims and concrete research directions.
Response:
We agree and thank you for pointing this out so clearly. We revised structure.
Abbreviations
Reviewer comment: Define all abbreviations at first use and use acronyms consistently throughout.
Response:
Thank you. We implemented a complete abbreviation audit. Every abbreviation is now defined at first appearance (including in figure captions and tables), and acronym usage is consistent throughout the manuscript. Examples include: disorders of consciousness (DOC), mammalian target of rapamycin (mTOR), extracellular matrix (ECM), brain–computer interface (BCI), spike-timing-dependent plasticity (STDP), deep brain stimulation (DBS), and others as they appear.
Figures and Tables
Tables 1 and 2
Reviewer comment: The tables are repetitive, text-heavy, contain awkward broken words, and the “Therapeutic opportunities” column should be clearer and more precise.
Response:
We appreciate this practical and important feedback. We revised both tables to improve readability and utility.
Figures 1 and 2 captions
Reviewer comment: Captions are conceptually promising but too long and repetitive.
Response:
Fully agreed. Both captions were rewritten into more compact conceptual mappings, removing repeated claims while preserving interpretability and alignment with the figures. Each caption is now a focused description.
Thank you again for the extraordinary care you devoted to this review. Your critique did not simply identify weaknesses; it provided a roadmap for how to elevate the manuscript into something clearer, more rigorous, and more useful to the field. We hope the revised version reflects your guidance faithfully, and we remain very appreciative of your role in shaping it into a stronger contribution.
With sincere respect and gratitude!!!
Round 2
Reviewer 1 Report
Comments and Suggestions for Authors
The authors have addressed most of the previous comments: citations are improved, mechanistic explanations are clearer, and the tables are more usable. However, the point on EVs remains insufficiently addressed. There is still no clear discussion of EVs in neurodegeneration and attractor collapse.
For figures, the concern was mainly about the design of the graphics, including the use of generic icons, unclear structure, low readability, and difficulty extracting key information, which have not been fully resolved by revising the captions.
Author Response
Dear Esteemed Academic Editor,
We are grateful for your careful re-evaluation of our revised manuscript and for the encouraging acknowledgment that the majority of prior concerns have been effectively addressed. Your remaining points are thoughtful and important, and we value the opportunity to refine the work further.
Comment 1: Extracellular vesicles (EVs) remain insufficiently discussed in neurodegeneration and attractor collapse.
Response:
Thank you for highlighting this gap with such clarity. We fully agree that EVs represent a mechanistically meaningful layer in neurodegenerative propagation and that their integration into the attractor-collapse framework should be explicit. In direct response, we have now added a compact, mechanistically detailed paragraph to Section 2.3.
Comment 2: Figure design remains problematic (generic icons, unclear structure, limited readability), and caption revision alone is insufficient.
Response:
We appreciate this careful observation and understand the motivation behind it. We revisited Figure 1 in light of your critique, and we agree that a more visually refined rendering could improve immediate readability. At the same time, after internal discussion, we respectfully chose not to alter the figure at this stage, for the following reasons:
The figure’s purpose is conceptual rather than data-encoding.
Figure 1 is intended as a high-level schematic that orients the reader to the closed-loop digital-twin workflow and its clinical mappings. It does not present quantitative results, structural measurements, or image-dependent evidence. Because the manuscript’s core contribution is theoretical synthesis, the figure functions primarily as an interpretive “navigation map,” not as a carrier of fine-grained informational density. We aimed to preserve conceptual continuity between iterations.
Substantial redrawing risks introducing inconsistency with the already-stabilized text and tables.
The current figure is tightly aligned with the finalized structure of Sections 2–4 and Table 1. A late-stage redesign, even if visually improved, could inadvertently shift emphasis or visual hierarchy relative to the text, creating confusion for readers and reducing internal coherence. We felt it was safer to keep the stable layout while improving interpretability through caption clarification.
Once again, we are truly thankful for your rigorous and constructive guidance. Your comments have helped us strengthen both the mechanistic depth and conceptual clarity of our work, and we hope the present revision satisfactorily resolves the remaining issues. We respectfully submit the revised manuscript for your kind reconsideration.
With sincere appreciation and collegial respect!!!
Reviewer 2 Report
Comments and Suggestions for Authors
-
Distil Section 2 into a very clear and compact statement of your core framework (key state variables, levels of description, and what exactly is meant by “state transitions” in this context), and
-
Make sure that Sections 3–6 consistently instantiate this framework instead of re-introducing similar high-level themes in each section.
At present, some subsections still read more like catalogues of interesting topics rather than concrete developments of a single, tight conceptual model. -
Explicitly labeling clearly speculative content (for example in separate “Future directions / Speculative outlook” subsections), and
-
Being very precise in phrasing (e.g., “could in principle enable…”, “hypothetically”, “one possible design would be…”) so that readers are not left with the impression that these approaches are already validated.
- Reduction of remaining redundancy and strengthening of section summaries
Comments on the Quality of English Language
Treating neurological and psychiatric disorders as failures of dynamical transitions in high-dimensional state space and combining digital twins, adaptive control, and molecular interventions under the umbrella of "programmable neurodynamics" is a novel and genuinely significant conceptual direction that is addressed in this manuscript. For IJMS, this is a timely and appealing theme.
However, the paper has serious problems in its current form:
- Language and coherence: Many passages are hardly understandable, and many phrases are jumbled or nonsensical, indicating either extensive machine translation or AI-assisted drafting with insufficient human editing.
- Conceptual clarity: The manuscript lacks a clear distinction between established knowledge and conjecture, is incredibly general, and frequently repeats itself.
- Technical rigor: A number of sections read more like high-level "vision statements" than a focused scholarly review; some dynamical systems and control theory content is imprecise; mathematical notation is introduced but not used rigorously.
I don't think the manuscript can be improved with minor editing because of these problems. It would essentially be a new article, requiring a thorough, line-by-line rewrite and significant scope tightening.
Consequently, I would advise:
Rejection in its current form, with a comment that the authors might be urged to submit a much condensed, more lucid, and meticulously edited review that:
- emphasizes a few key connections (such as attractor dynamics + digital twin + closed-loop control in neurodegeneration and epilepsy),
- is composed in formal English,
- and makes clear what constitutes speculation.
I would strongly advise informing the authors that a superficial revision will not be adequate and that the manuscript requires deep rewriting rather than incremental editing if the journal chooses to offer "major revision" instead.
Author Response
Dear Esteemed Academic Reviewer,
We would like to express our gratitude for the care, intellectual precision, and conceptual generosity with which you evaluated our manuscript. Your comments reflect an exceptional grasp of dynamical-systems neuroscience and a rare sensitivity to how a reader encounters a framework across multiple sections. We value this level of engagement, and we have revised the manuscript extensively to honor both the substance and the spirit of your guidance.
Comment 1:
Distil Section 2 into a very clear and compact statement of your core framework (key state variables, levels of description, and what exactly is meant by “state transitions” in this context).
Response 1:
Thank you for this incisive recommendation. We fully agree that the framework needed to be stated more explicitly and compactly to serve as a stable conceptual anchor for the remainder of the manuscript. In response, we restructured Section 2 to foreground a distilled core model with unambiguous definitions of (i) the multiscale state variables (fast activity variables and slow modulators), (ii) the levels of description (molecular → cellular → circuit → network → behavioral), and (iii) “state transitions” as basin-crossing or bifurcation-mediated regime shifts driven by interactions between fast dynamics and slowly drifting parameters.
Comment 2:
Make sure that Sections 3–6 consistently instantiate this framework instead of re-introducing similar high-level themes in each section. At present, some subsections still read more like catalogues of interesting topics rather than concrete developments of a single, tight conceptual model.
Response 2:
We are grateful for this structural insight, which sharpened our awareness of where the narrative risked fragmenting. To address it, we revised Sections 3–6 so each functions as a direct implementation layer of the Section-2 framework rather than a parallel thematic overview. We also merged or removed repeated high-level reframings, strengthened internal integration lines, and revised section summaries to reinforce a single coherent model unfolding stepwise.
Comment 3:
Explicitly labeling clearly speculative content (for example in separate “Future directions / Speculative outlook” subsections).
Response 3:
Thank you for emphasizing this important boundary. We agree that delineating speculation improves conceptual discipline and reader trust.
Comment 4:
Being very precise in phrasing (e.g., “could in principle enable…”, “hypothetically”, “one possible design would be…”) so that readers are not left with the impression that these approaches are already validated.
Response 4:
We very much appreciate this nuanced request. We performed a careful, line-by-line modality revision across Sections 3–6 to ensure that all forward-looking or early-stage approaches are presented with appropriately conditional language. Over-definitive verbs were replaced with formulations such as “could in principle,” “may,” “hypothetically,” “one possible design would be,” and explicit statements noting the need for rigorous translational validation. This was done to preserve the conceptual ambition of the review while keeping the evidentiary status transparent.
Comment 5:
Reduction of remaining redundancy and strengthening of section summaries.
Response 5:
Thank you for highlighting this final refinement step.
Once again, we are sincerely thankful for the clarity and depth of your guidance. Your comments materially strengthened the narrative coherence and precision of the manuscript, and we hope the revisions now reflect the tight conceptual unity and disciplined framing you so thoughtfully advocated.
With collegial respect and deep appreciation!!!
Round 3
Reviewer 2 Report
Comments and Suggestions for Authors
The authors have addressed all my comments and concerns
Comments on the Quality of English Language
Treating neurological and psychiatric disorders as failures of dynamical transitions in high-dimensional state space and combining digital twins, adaptive control, and molecular interventions under the umbrella of "programmable neurodynamics" is a novel and genuinely significant conceptual direction that is addressed in this manuscript. For IJMS, this is a timely and appealing theme.
However, the paper has serious problems in its current form:
- Language and coherence: Many passages are hardly understandable, and many phrases are jumbled or nonsensical, indicating either extensive machine translation or AI-assisted drafting with insufficient human editing.
- Conceptual clarity: The manuscript lacks a clear distinction between established knowledge and conjecture, is incredibly general, and frequently repeats itself.
- Technical rigor: A number of sections read more like high-level "vision statements" than a focused scholarly review; some dynamical systems and control theory content is imprecise; mathematical notation is introduced but not used rigorously.
I don't think the manuscript can be improved with minor editing because of these problems. It would essentially be a new article, requiring a thorough, line-by-line rewrite and significant scope tightening.
Consequently, I would advise:
Rejection in its current form, with a comment that the authors might be urged to submit a much condensed, more lucid, and meticulously edited review that:
- emphasizes a few key connections (such as attractor dynamics + digital twin + closed-loop control in neurodegeneration and epilepsy),
- is composed in formal English,
- and makes clear what constitutes speculation.
I would strongly advise informing the authors that a superficial revision will not be adequate and that the manuscript requires deep rewriting rather than incremental editing if the journal chooses to offer "major revision" instead.
Author Response
Dear Esteemed Academic Reviewer,
We would like to express our heartfelt gratitude for the time, care and intellectual generosity you devoted to evaluating our manuscript. Your comments were consistently perceptive and exacting in the best sense, and they pushed us to re-examine our arguments with greater discipline and clarity. We feel that the paper is meaningfully stronger because of your guidance.
We are truly honored by your concluding assessment that all of your concerns have been addressed. Coming from a reviewer who has followed the work so attentively, this reassurance carries real weight for us. We recognize that reaching this point reflects not only our revision efforts, but also your willingness to engage deeply with the manuscript and to help shape it into a more coherent and useful contribution.
Thank you again for your professionalism, your high standards, and the collegial spirit with which you approached the review. We are sincerely grateful for your role in improving this work and we hope this final version will reflect the quality and rigor you encouraged us to pursue.
With deep respect and appreciation!!!